# Thermodynamic Assessment of the Effects of Intermittent Fasting and Fatty Liver Disease Diets on Longevity

**DOI:** 10.3390/e25020227

**Published:** 2023-01-25

**Authors:** Melek Ece Öngel, Cennet Yildiz, Özge Başer, Bayram Yilmaz, Mustafa Özilgen

**Affiliations:** 1Department of Physiology, Faculty of Medicine, Yeditepe University, Kayısdagi, Atasehir, Istanbul 34755, Turkey; 2Department of Food Engineering, Yeditepe University, Kayısdagi, Atasehir, Istanbul 34755, Turkey

**Keywords:** intermittent fasting, diet therapy, non-alcoholic fatty liver disease, Child–Pugh Score, entropic age, metabolic lifespan entropy generation, longevity, second law of thermodynamics

## Abstract

Organisms uptake energy from their diet and maintain a highly organized structure by importing energy and exporting entropy. A fraction of the generated entropy is accumulated in their bodies, thus causing ageing. Hayflick’s entropic age concept suggests that the lifespan of organisms is determined by the amount of entropy they generate. Organisms die after reaching their lifespan entropy generation limit. On the basis of the lifespan entropy generation concept, this study suggests that an intermittent fasting diet, which means skipping some meals without increasing the calories uptake in the other courses, may increase longevity. More than 1.32 million people died in 2017 because of chronic liver diseases, and a quarter of the world’s population has non-alcoholic fatty liver disease. There are no specific dietary guidelines available for the treatment of non-alcoholic fatty liver diseases but shifting to a healthier diet is recommended as the primary treatment. A healthy obese person may generate 119.9 kJ/kg K per year of entropy and generate a total of 4796 kJ/kg K entropy in the first 40 years of life. If obese persons continue to consume the same diet, they may have 94 years of life expectancy. After age 40, Child–Pugh Score A, B, and C NAFLD patients may generate 126.2, 149.9, and 272.5 kJ/kg K year of entropy and have 92, 84, and 64 years of life expectancy, respectively. If they were to make a major recommended shift in their diet, the life expectancy of Child–Pugh Score A, B, and C patients may increase by 29, 32, and 43 years, respectively.

## 1. Introduction

### 1.1. Second Law Analysis Focusing the Organisms

Organisms live at far-from-equilibrium with their surroundings while maintaining homeostasis, importing exergy, and exporting entropy [1]. The early second law studies [2] aimed to achieve easy release of entropy from the human body; later these assessments aimed to evaluate human comfort at different temperature and humidity levels [3,4,5] and evaluate the effects of this process on each gender [6]. Later, these analyses aimed to understand sleeping comfort [7] why women live longer than men [8], assessment of the longevity of athletes [9], and the destructive effect of a disease on the health of the people [10]. These concepts were also employed to understand the changes occurring in the body during pregnancy and lactation [11] and during ageing [12]. Intermittent fasting (IF), skipping meals without increasing the calories of the others, is among the most popular contemporary weight loss diets. One-quarter of the world’s population has non-alcoholic fatty liver disease (NAFLD) [13], and more than 1.32 million people died in 2017 because of chronic liver diseases [14]. Feasible flux and metabolite activity profiles are determined by thermodynamics on the basis of a genome-scale metabolic flux analysis. This method involves the use of a set of linear thermodynamic constraints in addition to the mass balance constraints and produces flux distributions that do not contain any thermodynamically infeasible reactions or pathways [15]. Both IF and the treatment of NAFLD operate through thermodynamics-based genome-scale metabolic fluxes.

### 1.2. Intermittent Fasting

IF is recommended to treat diabetes and obesity [16], improve sleep quality and gut microbiota, decrease blood glucose levels, and facilitate weight loss [17]. When there is no food ingestion, the body begins to break down fat to produce fatty acids and glycerol and use them in energy metabolism. After 8–12 h of fasting, ketone bodies, such as acetoacetate, 3-beta-hydroxybutyrate, and acetone, are produced and then start to induce metabolic changes [18] and increase autophagy. Autophagy is a cellular housekeeping process that causes the degradation of damaged cytoplasmic components, such as misfolded proteins or damaged organelles. With ageing, cellular structure of the body begins to lose its ability to maintain and sustain the organismal development and reproduction [19,20,21,22], and compensatory systems decline over time. Metabolism responds to IF with better maintenance of its cells and recycling of the damaged molecules [23]. Autophagy can enhance longevity by decreasing inflammation, removing dysfunctional mitochondria, and reducing toxic proteins within lysosomes [24].

### 1.3. Chronic Liver Diseases

The occurrence of NAFLD tends to increase as people maintain a sedentary lifestyle and intake more calories than their metabolism can consume. In NAFLD, reduced glucose oxidation and increased fat oxidation may be observed, and basal metabolic rate may increase, but still, adequate energy may not be provided to the cells [25]. There are no specific dietary guidelines available for the treatment of NAFLD. Generally, weight loss and diet changes are recommended as the primary treatment [26]. Elevated fat intake with excessive amounts of n-6 fatty acids may play a role in promoting NASH (non-alcoholic steatohepatitis, inflammation of the liver with concurrent fat accumulation) [27]. The degree of the NAFLD disease is evaluated according to the Child–Pugh Score, which divides the patients into three categories: Child–Pugh A—people have normal hepatic conditions, Child–Pugh B—patients possess moderately weak hepatic function, and Child–Pugh C—patients have advanced hepatic dysfunction [28].

The American Heart Association [29] recommends a dietary pattern that provides not more than 5 to 6% of the calories from saturated fat. Zivkovic et al. [30] suggested increasing the intake of monounsaturated fatty acids (MUFA) instead of SFA (saturated fatty acids) and increasing the intake of complex carbohydrates instead of simple carbohydrates for NAFLD patients. Increasing consumption of n-3 fatty acids, found in fish oil and walnuts, may reduce liver damage in NAFLD patients. Consumption of excessive amounts of sucrose-sweetened soft drinks or fructose [30,31,32] may lead to the development of NAFLD. Low-carbohydrate, high-fat ketogenic diets may negatively affect metabolism and cause the development of NASH [7,31]. Asrih and Jornayvaz [33] suggested the presence of a link between the development of NAFLD and high-fat and ketogenic diets and recommended the consumption of low-refined carbohydrate, low-fat, and low-saturated-acid-containing diets. Non-digestible and fermentable fibre-containing, low-glycemic-index diets may help to improve the health of NASH patients.

Dietary Guidelines for Americans 2020–2025 [34] recommend a daily calorie intake range for healthy women and men of age 19 through 30 to be from 1800 to 2400 kcal and from 2400 to 3000 kcal, respectively. Such a diet should consist of 10–35% protein, 45–65% carbohydrate, and 20–35% fat. The amount of fibre in the diet should be adjusted to 14 g per 1000 calories. The percentage of saturated fat should be kept below 10% of total calories [34]. The Academy of Nutrition and Dietetics [34] recommends 500 to 750 calories/day of deficit for weight loss. The presence of saturated versus unsaturated fat and complex versus simple carbohydrates in a diet is reported to have opposing roles in the development of NAFLD [30,33]. The insulin sensitivity index and non-alcoholic steatohepatitis increase with the consumption of cholesterol and saturated fat and decrease with the consumption of polyunsaturated fat and fibre [35]. MUFA and polyunsaturated fatty acids (PUFA), particularly n-3 PUFA were reported to reduce liver fat content. In addition, it has been shown that MUFA can reduce body fat accumulation and impair plasma lipid levels, thus possibly preventing NAFLD development. A diet rich in n-3 PUFA has been found to reduce body weight and accumulation of hepatic triglycerides that causes liver steatosis [36].

Carbohydrates are stored in the body via conversion into fat or glycogen; NAFLD occurs since the fat accumulates in the liver, not elsewhere. Excessive fructose consumption in the diet may contribute to the development of NAFLD [30,33]. Zelber-Sagi et al. [37], when evaluating the diet consumed by the NAFLD patients, reported that they had consumed 50% more soft drinks, 27% more meat, and less omega-3-rich fish than the control group. Low carbohydrate and high-fat consumption by healthy individuals triggered insulin resistance [38]. The higher the damage to the liver, the less should be the conversion efficiency of the glycogen to the energy providing chemicals; therefore, energy conversion efficiency of glycogen should be worse in Child–Pugh Score C patients than that of Child–Pugh Score B patients, and that should be worse than Child–Pugh Score A patients.

Weight gain was prevented, but NAFLD and hepatic insulin resistance were observed in mice fed with a ketogenic diet [31]. Low-carbohydrate, high-fat ketogenic diets, in addition to being effective in weight loss, may negatively affect metabolism and have a role in the development of NASH [27].

### 1.4. Entropic Age

Organisms live at far-from-equilibrium with their surroundings while maintaining homeostasis, importing energy, and exergy and exporting entropy; a fraction of the generated entropy accumulates in the body and causes ageing [39]. Inactivation and malfunctioning of the biomolecules are among the consequences of ageing [40]. When organisms accumulate the maximum tolerable entropy, they die and then attain equilibrium with their environment [41]. Humans generate energy for their life processes through the consumption of food [42]. The catabolism of macronutrients, e.g., carbohydrates, fats, and proteins, into their small molecules leads to entropy generation [43]. Öngel et al. [10] calculated the nutrition and disease-related entropy generation on the basis of patient-specific diets and calculated the expected lifespan of the patient groups with different types of cancer. Silva and Annamalai [44] extended this concept and calculated entropy generation and its stress on different organs in the body. Then, Kuddusi [45] calculated the metabolic entropy generation rate as 0.46 × 10^−5^ kW/kg K and the expected lifespan of 79 years in the Marmara district of Turkey. The present study aims to evaluate the effects of intermittent fasting and fatty liver treatment diets on the longevity of patients.

## 2. Materials and Methods

A schematic description of the thermodynamic system boundaries uptakes and the metabolic waste of the dieting individual is presented in Figure 1.

### 2.1. Intermittent Fasting Diet Plans

In the present study, calculations are made for 25- and 50-year-old individuals with 170 cm of height and 80 kg of initial weight. Their basal metabolic rates (BMR) were estimated with the Harris–Benedict [46] equation (Table 1) as:(1)BMR=66.5+13.76W+5.003H−5.755A
where *W* represents the weight (kg), *H* is the height (cm), and *A* is the age of the subject. The daily caloric need of an individual is calculated as BMR × 1.2. With this approach, the caloric needs of 25-year-old and 50-year-old individuals were estimated as 2200 and 2000 kcal/day, respectively. Diet compositions suggested for these individuals are presented in Table 2. Amounts of the oxygen uptake and the metabolic waste for weight-maintaining, weight gain, and weight loss IF diets for the 25- and 50-year-old 80 kg individuals are presented in Table 3 for these individuals starting intermittent fasting to lose weight and get into a healthy body mass index (BMI) range. BMI is calculated by dividing the body mass of a person by the square of the body height and is expressed in units of kg/m². A person is classified as “overweight” with an initial BMI of (80/(1.70^2^)) = 28 kg/m^2^. This individual needs to decrease the BMI to 23 kg/m^2^ to get into the healthy weight range [47]; when the target BMI is attained, body weight becomes 66 kg, and then the BMR of the 25-year-old and the 50-year-old subjects become 1680 and 1537 W, respectively. One of the most common methods of IF is following the 16:8 plan. In this method, individuals eat for 8 h a day without any limitation, then do not consume any food, except tea, black coffee, and water for 16 h [16]. Equations (reactions) (2)–(4) indicate that O_2_ consumption must increase with the rate of metabolism of the nutrients. Table 3 indicates that O_2_ consumption increases in the case of the weight gain.

### 2.2. Fatty Liver Disease Diet Plans

A schematic drawing thermodynamic system describing the diet-induced fat accumulation or depletion in the liver is presented in Figure 2. In hepatic diseases, the risk of complications and mortality increase with malnutrition [48]; therefore, preventing malnutrition should be one of the main concerns. Even though NASH is commonly associated with obesity, it does not rule out the risk of malnutrition; furthermore, a sudden decrease in muscle mass in obese patients may be caused by sarcopenic obesity [49]. For obese patients, 5–10% weight loss is recommended, but protein intake must be monitored to prevent muscle mass loss [50]. To avoid hepatocyte necrosis (death of liver cells) and fibrosis from worsening, weight loss is recommended to be less than 1.6 kg per week [51]. Fibrosis may refer to the connective tissue deposition that occurs as part of normal healing or to the excess tissue deposition that occurs as a pathological process. An energy intake of 35–40 kcal/kg per day and a protein intake of 1.2–1.5 g/kg per day is recommended for NAFLD patients [52]. Including late-night snacks and breakfast in the diet might be beneficial in meeting the daily caloric needs [50]. In EASL [50] and ESPEN [52] guidelines, the importance of branched-chain amino acid (BCAA) consumption is highlighted; therefore, the majority of the protein intake should come from dairy protein sources. If the patient cannot uptake enough calories, leucine-enriched BCAA supplements might be used. The consumption of sugared beverages and desserts is strongly discouraged; in addition, sodium intake should be limited as well [52]. In the nutrition of liver patients, zinc and vitamin E supplementation were found highly beneficial [53]. In the present study, diets were planned for 100 kg obese patients (BMI > 30) with NAFLD. The caloric need of the patient with NAFLD with Child–Pugh Scores A, B, and C was assumed to be higher with advanced stage due to stress and catabolism process occurring in the body.

### 2.3. Entropy Generation and Lifespan Estimation

#### 2.3.1. Intermittent Fasting

Entropy generation rates for a 25-year-old overweight healthy person and a 50-year-old overweight healthy individual before and after the IF diet were calculated by using the thermodynamic properties presented in Table 4 by following the same procedure as Öngel et al. [8]. The model was designed by contemplating that 80 kg subjects from two different age groups consume the diets causing them to gain weight until they reach age 25 and 50, respectively, then they follow the IF diet to lose weight; after the weight loss, they consume the weight-maintaining IF diet. On the basis of these considerations, three different entropy generation rates were calculated depending on the prepared diets, and the lifespan of these subjects was estimated (Table 5).

It was assumed that a healthy person digests 92% of the proteins, 95% of the lipids, and 99% of the carbohydrates provided in the diet [54]. The amounts of these macronutrient intakes with the diets are presented in Table 6. In Equations (2)–(4), the metabolism of carbohydrates, lipids, and proteins was modelled in terms of glucose (C_6_H_12_O_6_), palmitic acid (C_6_H_32_O_2_), and an average of 20 amino acids (C_4.57_H_9.03_N_1.27_O_2.25_S_0.046_), respectively:C_6_H_12_O_6_ + 6O_2_ → 6H_2_O + 6CO_2_(2)
C_16_H_32_O_2_ + 23O_2_ → 16CO_2_ + 16H_2_O(3)
C_4.57_H_9.03_N_1.27_O_2.25_S_0.046_ + 4.75O_2_ → 3.245H_2_O + 3.935CO_2_ + 0.635CH_4_N_2_O(4)

Urine and feces are excreted after metabolism and absorption of nutrients. On the basis of the same procedure as Öngel et al. [10], the amount of urine was calculated to maintain the urea content as 20 g/L and excrete 65% of the urea out of the body. The amounts of the inhaled O_2_ and exhaled CO_2_ and H_2_O are shown in Table 7. Evaporative loss of H_2_O via perspiration is not considered in the calculations since water uptake does not contribute to entropy generation. It was assumed that the patient would drink more water in case of excessive perspiration, and it will not affect the results. Calculations of this study are based on the metabolic reactions presented in Equations (2)–(4). Carbon and hydrogen appear in all of these reactions and sulfur appears only in Equation (4), in the molecular formula of the “average of 20 amino acids”, where the ratio of S to C is 0.01, and the ratio of S to H is 0.005, indicating that the S compounds to entropy generation are extremely small; therefore, they are not considered in the calculations.

The heat generation rate by the body was calculated by using Equation (5), following the same procedure as explained by Ulu et al. [11]:(5)$$\Delta \dot{Q}=\sum \dot{n_{p}}{\left(h_{f}^{}\right)}_{p}-\sum \dot{n_{r}}{\left(h_{f}^{O}\right)}_{r}$$
where $\Delta \dot{Q}$ is the heat released by the body, $\dot{n_{p}}$ refers to the mole number rates of the products exiting the system, and $\dot{n_{r}}$ is the mole number rate of the reactants entering the system; $h_{f}^{O}$ represents the enthalpy of formation of the reactants uptaken at the standard conditions, and $h_{f}$ is the enthalpy of formation of the products of the metabolism leaving the body at 37 °C. Values of $h_{f}^{O}$ and $h_{f}$ are presented in Table 4 and adapted from Kuddusi [45]. According to Silva and Annamalai [55], the metabolism of one mole of each of glucose, palmitic acid, and the average 20 amino acids produces 32, 106, and 8 moles of ATP, respectively. The amount of heat released by their metabolism was found with Equation (6); metabolic efficiency (*η*) of glucose, palmitic acid, and the average of 20 amino acids were taken as 34.6%, 32.2%, and 10.4%, respectively [44].
(6)$$Q_{Sr}=\left(1-\eta \right)\times \Delta {\dot{H}}_{R}$$
where $\Delta {\dot{H}}_{R}$ is the enthalpy of metabolism. The entropy generation rates of the overweight 25- and 50-year-old subjects were computed by considering their diets via Equation (7) and are given in Table 5.
(7)$${\left(\sum ns\right)}_{out}-{\left(\sum ns\right)}_{in}-\sum \frac{Q}{T}=\Delta s_{gen}$$
where *n* is the mole number of chemicals taken in or out of the system, *s* represents the specific entropy of the chemicals, *T* describes the human body temperature (37 °C), and *Q* defines the heat generated by the body.

Silva and Annamalai [44] found that individuals generate 11,404 kJ/kg K during their whole life. On the basis of this principle, entropy generation rates at the ages of 25 and 50 were calculated, and then their remaining entropy generation rates were computed by adding entropy generation rates. After weight loss IF, it was assumed that longevities of the 25-year-old and 50-year-old subjects increase 10–25% owing to autophagy. Hence, it was assumed that annual entropy generation rates decreased in parallel with these values during the weight-maintaining diet (Table 5).

To perform calculations of the entropy generation rates during a weight loss IF diet and weight-maintaining intermittent fasting diet, the weights of 25- and 50-year-old individuals were determined with the help of the “Weight Loss Predictor Calculator” of Pennington Biomedical Research Center as 70.97 and 71.21 kg for each diet, respectively. The weight-versus-time plots are presented in Figure 3.

#### 2.3.2. Fatty Liver Disease

In the present study, the oxidation reaction of glucose, palmitic acid, and an average of 20 amino acids was chosen to represent the metabolism of carbohydrates, lipids, and protein, respectively, and calculations were performed on the basis of reactions Equations (2)–(4). All the calculations for NAFLD patients with different Child–Pugh Scores and healthy people were performed by using the data given in Table 4 based on the planned diet plans shown in Table 5. According to the procedure described by Öngel et al. [10], the entropy generation rate and lifespan estimation of the patients were calculated. It is assumed that oxidation of protein, fat, and carbohydrate is 28%, 46%, and 88% for the patient in Child–Pugh Score A, 31%, 55%, and 86% for the patients in Child–Pugh Score B, and 24%, 59%, and 88% for the patient with Child–Pugh Score C, respectively [56]. However, it was considered that during the disease, the patients in Child–Pugh Scores A, B, and C lost 5 kg, 10 kg, and 20 kg weight, respectively, due to complications. For a healthy person, these rates are 92%, 95%, and 99% for protein, fat, and carbohydrate, respectively [54]. On the basis of the entropy balance equation (Equation (7)) by Özilgen and Sorgüven [57] and Kuddusi [45], the entropy generation rate and lifespan estimation were found. It was assumed that $\frac{d{\left[m\;s\right]}_{system}}{dt}$ equals zero because the thermodynamic system, the body, is in a quasi-steady-state condition. To calculate the entropy generation rate during NAFLD (S_cirrhotic_), it was assumed that patients maintained a healthy life until they reached age 40 and consumed the diet of a healthy person, and then they became NAFLD patients; therefore, they started consuming a special diet for each Child–Pugh Score (Table 5). The lifespan entropy generation limit was 11,404 kJ/K kg in the calculations of Kuddusi [45]. The one-year survival rate is approximately 95%, 80%, and 44% for the patients with Child–Pugh Scores of A, B, and C, respectively [57]. Öngel et al. [10] estimated the disease-related entropy generation for 19 different varieties of cancer as:(Disease-related annual entropy generation rate by a patient) = (Total entropy generation by the patients in Δt years of remaining life span)/[(fraction of the patients surviving after Δt years) (Δt)].

The same expression is employed in this study for the NAFLD, and the remaining average lifetime (t_avg_) was estimated for each Child–Pugh Score patient and listed in Table 7. If persons are diagnosed with Child–Pugh Score C NAFLD at the age of 40, they would have already generated 4796 kJ/kg K entropy and may generate 6604 kJ/kg K of more entropy during the rest of their lifespan. Pinter et al. [58] indicated that 95%, 80%, and 44% of the NAFLD patients with Child–Pugh Score A, B, and C will survive the first year of the disease; therefore, when compared with the 119.9 kJ/kg K year of entropy generation rate of an obese (otherwise healthy) person, we may estimate that, on average, a Child–Pugh Score A patient may generate (119.9 kJ/kg K) [1/(0.95)]= 126.2 kJ/kg K of entropy annually. Similarly, Child–Pugh Score B and C patients may generate 149.9 and 272.5 kJ/kg K of entropy annually, respectively (Table 8).

## 3. Results and Discussions

### 3.1. Intermittent Fasting Diet

A weight-maintaining IF (16:8) diet reduced annual entropy generation rate to 28.3 kJ/kg K and 12.6 kJ/kg K for a 25-year-old individual and a 50-year-old individual in comparison with a diet causing weight gain, respectively. Therefore, the longevities of the 25-year-old and 50-year-old individuals are extended to 33 years and 14 years, respectively (Table 5). Total entropy generation until the age of 25 or 50 and the estimated lifespan of the subjects if they continue consuming the weight loss IF diet until the end of their lifespan are presented in Table 5, and the prevailing BMRs are presented in Table 5. As the age increases, the diet and the BMR decrease due to the decrease in calories in the diets.

Goodrick et al. [59] found that the IF diet led to an 83% extension of rat lifespan, and the ageing rate and mortality rate were decreased by the IF diet when compared with ad libitum. Ulu et al. [60] concluded that the human lifespan may increase by approximately 3% with the consumption of the IF diet for 30 days based on thermodynamic calculations. In the literature, there is no study regarding the effects of a long-term IF diet on human lifespan; because of this we could not compare our results with the experimental data showing how a long-term IF diet affects human lifespan extension. However, Yang et al. [61] found that the CR diet leads to a reduction of inflammation and upgrades metabolic homeostasis in humans and helps protect the body from the effects of ageing and slows the rate of the ageing process. Accumulation of certain proteins in the gut, muscle, and neuron cells accelerates ageing; therefore, autophagy appears as an important process and an extension of the lifespan and slows down the ageing process [62]. In a thermodynamic view, dysregulation of autophagy may increase the entropy generation rate [63].

In the present study, approximately 25% and 13.5% of calories were reduced in the weight loss and weight-maintaining IF diets in comparison with the weight gain diets for a 25-year-old individual. The proposed diet reduces these calories by approximately 25% and 15% for a 50-year-old individual. Our results show that the increase in the lifespan of the 25-year-old individual depending on IF diet was 19 years higher than that of the 50-year-old individual. Ravussin et al. [64] suggested that if an individual starts a 20% CR diet, implying reduction of the calories by 20%, at 25 years of age and maintains it until age 80, the lifespan of that person may increase by approximately 5 years. On the other hand, it extends it only two months if an individual follows a 30% CR diet at 55 age and sustains it for 22 years. If the Rhesus monkeys start consuming an early CR diet, they may live longer than the old-age onset [65]. The results of our calculations are compatible with those of the previous reports. Willcox et al. [66] argued that the Okinawan diet style increases their 65 years of lifespan from 6% (1.3 years) to 20% (3.6 years) when compared with Japanese and American lifespans, respectively. Okinawans’ survival rate did not show an increase only due to CR, but it also depended on diet composition. Yildiz et al. [67] found that more calorie intake changed gut microbiota composition negatively; hence, it caused more entropy generation when compared with the same composition of low-calorie diet consumption. Moreover, Öngel et al. [8] determined that various diet types affected the human lifespan, leading to the generation of entropy in different amounts, and the Mediterranean diet increased the human lifespan, causing a lower level of entropy generation. More protein and fat intake causes a much higher entropy generation rate, and, therefore, lifespan expectancy decreases [10,44].

During the IF diet period, the oxygen consumption rate of individuals decreases when compared with the diet causing weight gain (Table 3). It was also found that CR diminished the level of oxidative damage and failure of function related to oxidative damage. Hershey and Lee [41] argued that entropy generation increases in parallel with the basal metabolic rate (BMR) and slows with age; prolonged overfeeding causes an increase in BMR and, hence, the s_gen_. Our results show that the increase in BMR and entropy generation rate parallels, and BMR decreases, with weight loss after the IF diets (Table 1). Reduced oxidation energy implies reduced entropy generation.

### 3.2. NAFLD Diets

Çatak et al. [68], by using the lifespan entropy generation method, estimated that the longevity of an obese person who uptakes 10% more nutrients than a normal person is approximately 5 years less than that of a non-obese person. The diet lists were prepared for a healthy obese person to consume before developing NAFLD and for the patients with each Child–Pugh Score of NAFLD. On the basis of these diets, we calculated patients’ entropy generation rates and lifespans during their healthy life and NAFLD by following the procedure of Öngel et al. [8]. Thermodynamic properties of the macronutrients and their oxidation products, which are employed in these calculations, are presented in Table 4. According to Table 8, a healthy obese person generates annually 119.9 kJ/kg K per year and 4796 kJ/kg K of entropy during their 40 years of life. If these individuals maintain a healthy life and consume the same diet, they may live for 94 years. The risk of steatosis development increases depending on BMI. The incidence of steatosis is 65% in obese people with Child–Pugh Scores of A and B [69]. Here, we analyzed the effects of diets prepared for each Child–Pugh Score patient’s entropy generation rate and lifespan. Our results showed that a Child–Pugh Score A NAFLD patient generates 126.2 kJ/kg K of entropy and will have 92 years of lifespan. Therefore, it may be assumed that NAFLD with Child–Pugh Score A shortened the lifetime of an obese individual by 2 years. As a result of the calculations, it was found that the NAFLD patients in Child–Pugh Score B may live 84 years by generating annually 149.9 kJ/kg K entropy (Table 7). It can be estimated that the longevity of the obese patient in Child–Pugh Score B can decrease by 10 years. We found that patients with Child–Pugh Score C may generate 272.5 kJ/kg K year of entropy and have the shortest lifespan (64 years), and their life expectancy will be 30 years less when compared with the lifespan estimation of a healthy obese person.

Child–Pugh Score A, B, and C patients have good, mid, and poor nutritional status, respectively. In the patient with Child–Pugh Score C, ascites and encephalopathy develop, and they are at advanced and refractory levels. They are absent in a patient with Child–Pugh Score A and minimal levels in a patient with Child–Pugh Score B [70]. Our results showed that the patient’s lifespan with Child–Pugh Score C shortens by 26 and 15 years compared with the patients with Child–Pugh Scores A and B, respectively.

Schneeweiss et al. [71] suggested that this value and the basal metabolic rate increased in parallel with Child–Pugh Scores, e.g., the oxygen consumption rate of the whole body of the NAFLD patients was (228.8 ± 7.1 mL/min)/1.73 m^3^ although it was (206.5 ± 4.0 mL)/min/1.73 m^3^ in normal individuals. The abnormal increase in oxygen consumption causes much more entropy generation. Therefore, the disorder of the liver and, finally, the whole body increases, and ageing occurs and lifespan shortens. Our results also show that oxygen consumption increases with the stage of the NAFLD, and it was the highest in the patient at Child–Pugh Score C (Table 7).

This study is based on the concepts originally suggested by Silva and Annamalai [55,72] and Annamalai and Silva [44]. Recently Annamalai [73] argued the similarity between oxygen-deficient combustion and metabolism and explained why proteins have low metabolic efficiency (η = 10%) and cause higher entropy generation when compared with carbohydrates and fats (almost η = 40). Proteins are used mainly for bodybuilding or replacing cells. In the present study, we have extended this concept to diet treatment with IF and treatment of the liver diseases. Our results pertinent to IF diet are in agreement with the findings of Siclair [74], who argue that CR is not simply a passive effect but an active, highly conserved stress response that evolved early in life’s history to increase an organism’s chance of surviving adversity. Öngel et al. [8] presented the life expectancy of the women with a telomere-length-regulated and diet-based entropic assessment on Mediterranean, Western (American), ketogenic, and vegan diets based on the methods presented by Hayflick [40], Annamalai and Silva [44], and Silva and Annamalai [66]. With all of these diets, lifespan estimations of the women were longer than those of the men. Faster shortening of the telomere lengths in men was the major reason for the shorter life expectancy. The highest and the lowest life expectancy for women were estimated with Mediterranean and the vegetarian diets, respectively; men were estimated to have the longest life span with the vegetarian diet and the shortest life span with the ketogenic diet. In the present study, the calculation presented by Öngel et al. [8] was not repeated to avoid redundancy.

## 4. Conclusions

In the present study, diet lists were prepared for a healthy obese person and Child–Pugh Score of A, B, and C for patients who developed NAFLD after age 40. It was found that healthy obese people generate 119.9 kJ/kg K per year of entropy and, in total, 4796 kJ/kg K entropy in their 40 years of life. If a healthy obese person continues to consume the same diet after the age of 40, this person may have 94 years of life expectancy. Meanwhile, whole body of the Child–Pugh Score A, B, and C NAFLD patients may generate 126.2, 149.9, and 272.5 kJ/kg K year of entropy, and they would have 92, 84, and 64 years of life expectancy, respectively. These results imply that if the patients were to continue to consume their present diets, the NAFLD may decrease the lifespan expectancy of Child–Pugh Score A, B, and C patients by 2, 10, and 30 years, respectively. However, if they were to make a major shift, as recommended in the present study, the lifespan expectancy of Child–Pugh Score A, B, and C patients may increase by 29, 32, and 43 years, respectively. This study was carried out to demonstrate the effects of diets on the lifespan expectancy of NAFLD patients. It should be emphasized that the predictions given here are based on the assumption that the subjects will not develop any other diseases or health complications other than the NAFLD in their lifespans. In the case of such other complications, we would not expect these results to be valid.

## Figures and Tables

**Figure 1 entropy-25-00227-f001:**
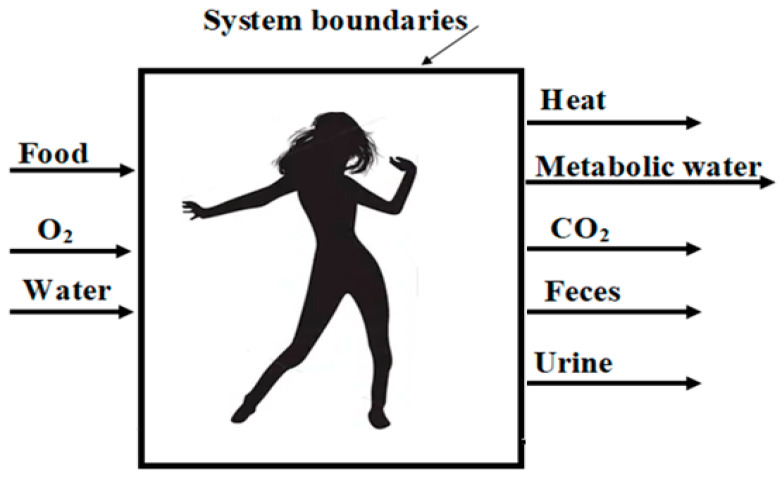
Schematic description of the thermodynamic system boundaries uptakes and the metabolic waste of the dieting individual.

**Figure 2 entropy-25-00227-f002:**
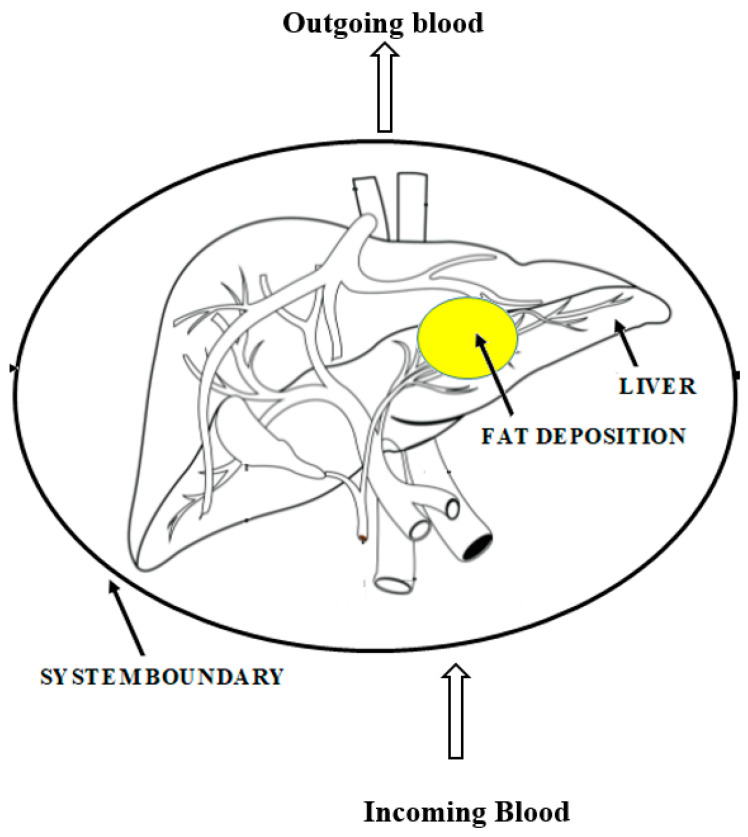
The thermodynamic system describing the fat accumulation in the liver.

**Figure 3 entropy-25-00227-f003:**
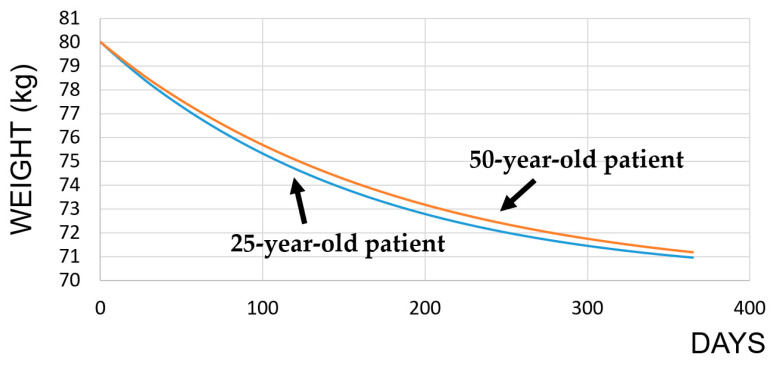
Weight of the individuals on the IF diet.

**Table 1 entropy-25-00227-t001:** Estimated BMR for each person (calculated with the BMR of the calculator net, available at https://www.calculator.net/about-us.html; accessed on 30 December 2022).

Subjects	Initial BMR	After IF BMR
25-year-old men	1885 kcal/day	1680 kcal/day
	7879 kJ/day	7022 kJ/day
50-year-old men	1618 kcal/day	1478 kcal/day
	6763 kJ/day	6178 kcal/day

**Table 2 entropy-25-00227-t002:** The estimated composition of each diet.

Type of Diet	Calorie (kcal)	Carbohydrates (g)	Protein (g)	Lipid (g)
Weight-maintaining IF diet for a 25-year-old, 80 kg person	2210	318	99	59
Weight-maintaining IF diet for a 50-year-old, 80 kg person	2010	295	86	54
Weight gain IF diet for a 25-year-old, 80 kg person	1912	276	94	48
Weight gain IF diet for a 50-year-old, 80 kg person	1711	257	74	43
Weight loss IF diet for a 25-year-old, 80 kg person	1663	242	68	47
Weight loss IF diet for a 50-year-old, 80 kg person	1510	225	67	38

**Table 3 entropy-25-00227-t003:** Amounts of the oxygen uptake and the metabolic waste for the weight-maintaining, weight gain, and weight loss IF diets for the 25- and 50-year-old, 80 kg individuals.

	O_2_ (g/day)	H_2_O (g/day)	CO_2_ (g/day)	Urine (g/day)	Dry Feces (g/day)
Weight-maintaining IF diet for a 25-year-old, 80 kg person	532.9	257.6	652.0	894.7	1646.5
Weight-maintaining IF diet for a 50-year-old, 80 kg person	475.7	232.0	584.6	677.0	1856.1
Weight gain diet for a 25-year-old, 80 kg person	613.2	296.6	748.4	942.3	1601.8
Weight gain diet for a 50-year-old, 80 kg person	560.0	271.7	684.6	786.8	1751.9
Weight loss IF diet for a 25-year-old, 80 kg person	463.7	224.7	565.2	647.2	1883.6
Weight loss IF diet for a 50-year-old, 80 kg person	420.0	204.5	515.7	613.0	19,016.9

**Table 4 entropy-25-00227-t004:** Thermodynamic properties of the macronutrients and the products of their oxidation at 1 atm (adapted from Kuddusi [45]).

Chemical	Enthalpies (h) and Entropies of the Macronutrients (s) of O_2_, CO_2_, and H_2_O at 1 atm and 298 K and 310 K			
	$h_{298\;K}$ (kJ/kmol)	s at (298 K) (kJ/kmol K)	$h_{9310\;K}$ (kJ/kmol)	s at (310 K) (kJ/kmol K)
C_6_H_12_O_6_ (glucose)	−1260 × 10^3^	212	-	
C_16_H_32_O_2_ (palmitic acid)	−835 × 10^3^	452	-	
C_4.57_H_9.03_N_1.27_O_2.25_S_0.046_ (average of the 20 amino acids)	−385 × 10^3^	1.401 × 119	-	
O_2_	8682	218		220
H_2_O			10,302	219
CO_2_			9807	243

**Table 5 entropy-25-00227-t005:** Entropy generation rates and lifespan estimation by 25- and 50-year-old individuals depending on the diets.

	25-Year-Old Individual	50-Year-Old Individual
Total entropy generation until the age of 25 or 50 and BMR	2560 kJ/kg K 7879 kJ/day	4651 kJ/kg K 6763 kJ/day
Annual entropy generation due to consumption of the weight gain IF diet (kJ/kg K year)	102.4	135
Annual entropy generation rate due to IF weight-maintaining diet (kJ/kg K year)	81.1	121
Annual entropy generation rate due to weight loss IF diet (kJ/kg K year)	74.1	91
Estimated lifespan (years) when they continue consuming the IF weight gain diet until the end of their lifespan	110	100
Estimated lifespan (years) when they continue consuming IF weight-maintaining diet until the end of their lifespan	133	106
Estimated lifespan of the subjects if they continue consuming the weight loss IF diet until the end of their lifespan	143 years 7022 kJ/day	135 years 6178 kcal/day

**Table 6 entropy-25-00227-t006:** Diet plans for obesity-induced NFLD patients based on Child–Pugh Score and healthy obese person.

Child–Pugh Scores	Child–Pugh Score A	Child–Pugh Score B	Child–Pugh Score C	Healthy Obese Person
kcal/day	3000	3200	3300	3100
Carbohydrate (g/day)	428	416	470	430
Protein (g/day)	130	160	165	150
Fat (g/day)	92	110	79	98
Total (g)	650	686	714	687

**Table 7 entropy-25-00227-t007:** The amount of the consumed oxygen, exhaled carbon dioxide, metabolic water, and the excreted waste.

	Healthy Obese Person	Child–Pugh Score A	Child–Pugh Score B	Child–Pugh Score C
O_2_ (g/day)	898	574	619	626
H_2_O (g/day)	428	294	307	320
CO_2_ (g/day)	1081	728	763	793
Dry feces (g/day)	1078	2326	2221	233
Urine (g/day)	1428	377	513	394

**Table 8 entropy-25-00227-t008:** Entropy generation rates and lifespan estimations for healthy obese and Child–Pugh Score A, B, and C individuals.

	Healthy Obese Person	Child–Pugh Score A Patient	Child–Pugh Score B Patient	Child–Pugh Score C Patient
The annual entropy generation rate ${\dot{S}}_{gen}$ until age 40 (kJ/kg K year)	119.9	119.9	119.9	119.9
Total entropy generation in 40 years S_healthy_ (kJ/kg K)	4796	4796	4796	4796
The annual entropy generation rate ${\dot{S}}_{gen}$ until age 40 in the case of no diet change (kJ/kg K year)	119.9	126.2	149.9	272.5
The annual entropy generation rate ${\dot{S}}_{gen}$ after the age of 40 in the case of a change of diet (kJ/kg K year)	122	76.4	87.2	98.8
Expected lifespan in the case of no diet change (years)	95	92	84	64
Expected lifespan in the case of the change of the diet (years)	-	127	116	107

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
