# Peer review of "Thermodynamic Assessment of the Effects of Intermittent Fasting and Fatty Liver Disease Diets on Longevity"

_entropy, 2023, doi:10.3390/e25020227_

Round 1

Reviewer 1 Report

Thermodynamic assessment of the effects of intermittent fasting and fatty liver disease diets on longevity

By Öngela et al

The  present manuscript  applies  the 2nd Law concept on life span {Silva and Annamalai, , Entropy Journal, 2012)  when calorie restrictions (CR)/intermittent fasting (IF)  recommended for non-alcoholic fatty liver disease (NAFLD-A, B and C) patients  where A, B and C are  a measure of severity with A being mild ( low score) and C being severe ( high score). It is well known that CR in general leads to longer life span and authors attempt to quantify using entropy generation (Sgen) or 2nd law concept  for  NAFLD patients. Thus, weight loss therapy through intermittent fasting (IF) seems to have favorable effects on body weight (BW) and relevant indicators of NAFLD .

The topic is  interesting. Extensive references are provide.  However more clarity and re-organization are required.  Thus, the manuscript in current form requires major revisions and they are summarized under major comments.

Major Comments

1.    The topic is of inter-disciplinary nature and hence requires a rudimentary approach to the topic so that readers in biology and thermodynamics can understand the concepts.

2.    Several medical terms used in manuscript were not explained. Thus, readers  with expertise in thermo. will have "rough" time following manuscript. Include a glossary of medical terms; NAFLD, NAFAD, Ubiquitination, cirrhosis, steatosis, autophagy  etc. since those in thermodynamics may not be familiar . Example NAFLD : “The Child-Pugh scoring system (also known as the Child-Pugh-Turcotte score, 1964, based on laboratory results for : serum bilirubin, serum albumin, ascites, neurological disorder, and clinical nutrition status.) predicts mortality in cirrhosis patients to guide the selection of patients who would benefit from elective surgery for portal decompression.  A - good hepatic function, B - moderately impaired hepatic function, and C - advanced hepatic dysfunction. , based on the scoring system was modified later by Pugh et al., substituting prothrombin time for clinical nutrition . status.” [https://www.ncbi.nlm.nih.gov/books/NBK542308/12/22/2022]. After  30 h fasting, stored glycogen in the liver is depleted and gluconeogenesis causes  the generation of glucose as a fuel for other tissues.

3.    All acronyms mut be listed under nomenclature. Missing NAFAD : Nonalcoholic fatty acid disease ? same as NAFLD?  Except that  NAFAD applies to liver?

4.    There are extensive studies by Sinclair group at Harvard on the effects of CR only lifespan. According to this group,  if caloricsummarize  intake is reduced by 30-40% lifespan is extended by 33 % { mice, rats, and monkeys}. Briefly their literature; may be appropriate at lines 331, 369, 372.  It seems that mitochondria die out  with age while CR revives them.

5.    Present BMI for 25 and 50 years in Table 1 along with: BMR for obese, obese and healthy males. Weight for  a) Normal (80 kg?), b) Obese (100 kg?) , c) Obese but healthy (?? Kg), d) NAFLD A, B and C. Table 1 and 2 must have  BMR, Calories need for Obese patients (100 kg) also in addition to normal (80 kg).

6.    Fatty liver: Is glycogen conversion to CH  limited due to NAFLD ? if so, present % conversion for A, B and C . I expect C to be the worst.

7.    CR generally results in weight loss.  Table  2 mentions  a) regular diet b) weight loss with  IF diet etc. and Table 3 mentions  c) weight gain diet, ( not included in Table 2) d) weight maintaining IF ( intermittent fasting) diet…Table 6 must include all above diets  ( a, b, c, d, etc.) before Table 3. Weight maintenance is  same as regular diet (see Table 2)? Include this diet. Qualitative figures may be presented on weight vs number of days for all above diets (somewhat like Fig. 2) and qualitative sgen vs year for normal, and obese healthy, and finally NAFLD A, B and C with or without diet restriction 

8.    Thermodynamic assessment deals  mostly with effects of CR on weight loss and reduced Calories on life span. But less on fatty liver disease itself  except that IF is required to reduce fatty liver. No results are presented how much liver fat  is reduced.  If data available, present them.

9.    No figure is presented for weight gain vs number of days. Include it in fig. 2

10. Lines 406-410: Authors talk about normal and obese persons who consume 10 % more nutrition. Same % in Calories? Mention this in earlier part of manuscript

11. Non-obese person lives 5 years longer based on sgen ? Present diet for non-obese vs obese.

12. Line 312; why 119.9/0.95;faster entropy generation why?  Estimation of sgen as 119.9/0.44= 272.5 kJ/kg K  for NAFLD C (since 44% NAFLD C will survive first year and 56 % die).Any rational explanation? Why not 119.9/0.56 ?

13. One of the the problems with CR is lack of enough ATP  per unit volume in performing life sustaining functions: heartbeat, respiration, etc. The 2nd law approach (which includes effect of met. efficiency)  rather than First law approach of Rubner is preferable. Include brief discussion.

14. Line 361: -25%, (loss in weight?) -13.5 % ( weight maintaining?)? reduction in Calories within a year after IF diet? ( Loss in weight) . Not clear.

15. Line 365: 20 % CR diet : means 20 % reduction in Calories? CR: Calorie restriction ? or Calorie Reduction?

16. Line 373: See Comment 7. Include separate table for Okinawan diet, Mediterranean Diet (MD) and compare with other diets . Line 379: How MD reduces sgen. Explain in terms of CH, F and P contents

17. Line 385: Calorie reduction must accompany with reduction in O2 consumption since energy release per unit mass of O2 remains fixed/. Thus, CR reduces O2  consumption or reduced amount of electron transfer {see Annamalai 2021, Systems;  also see Popovic, Heliyon, 2019] causing less damage to cells due to reduced “ions” or reduced oxidation damage. See lines 386-389 in Text and  link with reduced electron transfer under CR. One may quantify % reduction in electron transfer with % reduction in O2 consumed.

18. Line 399: reduced autophagy { when body does not have enough CH, fatty acids break down to ketones; abnormal cells are destroyed?) is only speculation not supported with data in the current manuscript. If there is supporting data from other literature cite them.

19. List sgen at 40 years for Normal male, Obese male , obese  healthy male ( 119.8 kJ/kg K? NAFLD A, B and C  in a table : explain in text how the number was  obtained

20. Are proteins included in Calories and sgen estimation? Urea : CH4N2O; atom balance yields 0.635 moles per empirical  mole of protein; did authors differentiate Urine vs Urea? Human urine : water, urea, (CH4N2O) inorganic salts, creatinine, ammonia etc.

21. Eq. 4 does not contain S balance? SO3 missing in Eq (4)

22. Also Eq. 3:  Correct it . fat : C16 H32 O2

23. Urea in urine: 20 g/L? if 1 L of urine = 1000 g, 20*100/1000 = 2 %? So, urea is only 2% of Urine?

24. Does Eq (5) includes evaporation loss of H2O?

25. Table 7: indicate below : O2g/day inhaled or consumed ?

26. Feces: typically, 75 % water + 25 % solid; 1078 g/day ?  dry or wet?

27. Weight change fig 2  after line 272. Include also weight gain for 25 and 50 yr.  as another plot.

28. Fig.2. If simple equation is available include this equation under caption for weight vs days .Cite ref. where equations are available for weight.  Caption: it is  not weight change but weight vs days after CR. Days counted  after 40 years or after IF start day

29. Clarify NAFLD: oxidation is only  28 % of protein intake , 46 % of fat intake, 88% of CH ….for NAFLD-A? so 12 % of CH is not oxidized? So O2 consumed decreases?

30. So, model up to 40 years; then special diet or IF diet for how long ? rest of life span?

31. 287-291:  weight loss over how many days? Then” these rates are 92, 95..… %..” Not clear

32. How is autophagy related to sgen? autophagy may require ATP  to destroy abnormal cells. How metabolic efficiency affects autophagy?

33. 384: O2 consumption increases with metabolic rate. Does weight gain or weight loss  relate to O2 consumed?

34. 386-389: important. More the O2 more electrons transfer more damage

35. Line 395. Present  a graph for sgen/BMR vs age. Does EEP decrease with age? Why reduced BMR?

Minor Comments

36. Table  4:  check entropy of protein. It must be  398 kJ per empirical mole per K

37. Table 5: refer weight loss, gain diet etc. to Table 2

38. Table 6: For CH, F and P, include kcal/day within parenthesis. E.g., 438 (kcal/day)…

39. Line 270: 70.97, 71.21 refer to weight not weight loss.

40. Table 7 caption: O2 not inhaled but consumed for metabolism

41. Line 266: reduced oxidation energy implies reduced entropy generation

42. 285: incomplete tot improper sentence. Confusing

43. 289:weight loss over how many days?

44. Lines 361-363 confusing

45. 390: EEP. Define. Include in acronym

46. Line 392: space missing

47. Section 3.2 may be moved earlier to explain NAFLD

48. Line 440; sentence awkward. Not clear.

49. Line 441-443: whole body ? or Liver only?

50. Line 447-449: Table 7 or Table 5?

51. Line 460-461: sgen whole body or liver?

52. 464-467: should be moved to Introduction section

Author Response

Thermodynamic assessment of the effects of intermittent fasting and fatty liver disease diets on longevity

Melek Ece Öngela, Cennet Yildizb, Özge Başer a, Bayram Yilmaza and Mustafa Özilgenb

a Yeditepe University, Faculty of Medicine, Department of Physiology, 34755, Kayısdagi, Atasehir, Istanbul Turkey

b Department of Food Engineering, Yeditepe University, 34755, Kayısdagi, Atasehir, Istanbul, Turkey

e- mail addresses and ORCID identifiers of the authors:

Melek Ece Öngel, meceongel@gmail.com; 0000-0003-4164-2213

Cennet Yildiz, cennet.yildiz@yeditepe.edu.tr; 0000-0002-5902-3608

Özge Başer, ozgebiyo@gmail.com; 0000-0003-0927-8247

Bayram Yilmaz Co-corresponding author), byilmaz@yeditepe.edu.tr; 0000-0002-2674-6535

Mustafa Özilgen (Co-corresponding author), mozilgen@yeditepe.edu.tr; 0000-0003-0522-3644

ABSTRACT

Organisms uptake energy from their diet and maintain a highly organized structure by importing energy and exporting entropy. A fraction of the generated entropy is accumulated in their bodies, thus it causes ageing.  Hayflick’s entropic age concept suggests that the lifespan of organisms is determined by the amount of entropy they generate. Organisms die after reaching their lifespan entropy generation limit. Based on the lifespan entropy generation concept, this study suggests that an intermittent fasting diet, which means skipping some meals without increasing the calories uptake in the other courses, may increase longevity. More than 1.32 million people died in 2017 because of chronic liver diseases, and a quarter of the world’s population has non-alcoholic fatty liver disease. There are no specific dietary guidelines available for the treatment of non-alcoholic fatty liver diseases, but shifting to a healthier diet is recommended as the primary treatment. A healthy obese person may generate 119.9 kJ/kg K per year of entropy and generate a total of 4,796 kJ/kg K entropy in the first 40 years of life. If this obese person continues to consume the same diet may have 94 years of life expectancy. After age 40, Child-Pugh Score A, B and C NAFLD patients may generate 126.2, 149.9 and 272.5 kJ/kg K year of entropy and have 92, 84 and 64 years of life expectancy, respectively. If they should make a major recommended shift in their diet, the life expectancy of  Child-Pugh Score A, B and C patients may increase by 29, 32 and 43 years, respectively.

Keywords: Intermittent fasting; diet therapy; non-alcoholic fatty liver disease; Child-Pugh Score; entropic age; metabolic lifespan entropy generation; longevity; second law of thermodynamics.

Biographical information:

Melek Ece Öngel has BS and MS degrees from the Department of Nutrition and Dietetics of the Yeditepe University and studying towards PhD degree in the department of Physiology in the Faculty of Medicine of the same university. She specializes in patient-tailored nutritional therapy, with an emphasis on its thermodynamic aspects.

Cennet Yildiz is a Food Engineer and a PhD candidate in Biotechnology Program at Yeditepe University in Istanbul, Turkey. She holds a BS in Food Engineering from Cumhuriyet University in Sivas, Turkey and MS in Chemical Engineering from Yeditepe University, in Istanbul, Turkey. Her current research interests focus on thermodynamic aspects of biological processes. She is currently studying at the Technical University of Munich, Olympiapark, Germany as a visiting student and carrying out part of her PhD work.

Özge Başer is a PhD candidate in the physiology program at Yeditepe University (in Istanbul, Turkey). She has a biology bachelor's degree from Hacettepe University (in Ankara, Turkey) and Physiology Master's degree from Yeditepe University. She is interested in metabolic disorders and neurodegenerative disease from a neuroscience perspective. She focuses on revealing the neural changes and hormonal regulation behind obesity as her thesis topic by performing brain surgery and behavioural tests on mice at Yılmaz Lab at Yeditepe University. She has currently a joint project with Linköping University (Sweden) by performing on Alzheimer disease model. Additionally, she is also interested in the ageing processes in the brain.

Bayram Yilmaz is a Professor of Medical Physiology at Yeditepe University in Turkey. He has a Doctor of Veterinary Medicine degree from Firat University (Turkey) and a PhD degree in Medical Physiology from the University of Glasgow (Scotland). He worked as a research assistant professor at the State University of New York at Albany and visiting scientist at the King’s College London and the University of Oxford. He is the author of seven book chapters and 140 refereed articles.

Mustafa Özilgen is a Chemical Engineer and a Professor at Yeditepe University in Turkey. He has received his BS and MS from Middle East Technical University, Turkey and PhD from the University of California at Davis, USA. He taught classes at the Middle East Technical University, Massey University, New Zealand and the University of California at Davis, USA, and worked as a consultant at Marmara Research Center, Turkey. He is the author of five books and 125 refereed articles. His current research interests focus on biothermodynamics, bioenergetics, kinetic aspects of cellular systems and environmental friendly food and bioprocessing.

  1. INTRODUCTION

1.1. Second law analysis focusing the organisms

Organisms live at far-from-equilibrium with their surroundings while maintaining homeostasis, importing exergy and exporting entropy  [1]. The early second law studies [2] were aiming to achieve easy release of entropy from the human body; later these assessments aimed to evaluate human comfort at different temperature and humidity levels [3,4] and evaluate the effects of this process on each gender [5]. Later, these analyses aimed to understand sleeping comfort [6], and why women live longer than men do [7] and assessment of the longevity of athletes [8] and the destructive effect of a disease on the health of the people [9]. These concepts were also employed to understand the changes occurring in the body during pregnancy and lactation [10] and during ageing [11]. Intermittent fasting (IF) means skipping meals without increasing the calories of the others is among the most popular contemporary weight loss diets. A quarter of the world’s population has a non-alcoholic fatty liver disease (NAFLD) [12], and more than 1.32 million people died in 2017 because of chronic liver diseases [13]. Feasible flux and metabolite activity profiles are determined by thermodynamics based on a genome-scale metabolic flux analysis. This method involves the use of a set of linear thermodynamic constraints in addition to the mass balance constraints and produces flux distributions that do not contain any thermodynamically infeasible reactions or pathways [14]. Both IF and the treatment of NAFLD operate through thermodynamics-based genome-scale metabolic fluxes.

1.2. Intermittent Fasting

IF is recommended to treat diabetes and obesity [15], improve sleep quality and gut microbiota, decrease blood glucose levels and facilitate weight loss [16]. When there is no food ingestion the body begins to break down fat to produce fatty acids and glycerol and use them in energy metabolism. After 8-12 hours of fasting ketone bodies, such as acetoacetate, 3-beta-hydroxybutyrate and acetone are produced and then start to induce metabolic changes [17], and increase autophagy. Autophagy is a cellular housekeeping process that causes the degradation of damaged cytoplasmic components such as misfolded proteins or damaged organelles. Metabolism responds to IF with better maintenance of its cells and recycling of the damaged molecules [18]. Autophagy can enhance longevity by decreasing inflammation, removing dysfunctional mitochondria and reducing toxic proteins within lysosomes [19].

  • Chronic liver diseases

The occurrence of NAFLD tends to increase as people maintain a sedentary lifestyle and intake more calories than their metabolism can consume. In NAFLD reduced glucose oxidation and increased fat oxidation may be observed, and basal metabolic rate may increase, but still, adequate energy may not be provided to the cells [20]. There are no specific dietary guidelines available for the treatment of NAFLD. Generally, weight loss and diet changes are recommended as the primary treatment [21]. Elevated fat intake with excessive amounts of n-6 fatty acids may play a role in promoting NASH (non-alcoholic steatohepatitis, inflammation of the liver with concurrent fat accumulation) [22]. The degree of the NAFLD disease is evaluated according to the Child-Pugh Score, which divides the patients into three categories: Child-Pugh A people have normal hepatic conditions, Child-Pugh B patients possess moderately weak hepatic function and Child-Pugh C patients have advanced hepatic dysfunction [23].

The American Heart Association [24] recommends a dietary pattern that provides not more than 5 to 6% of the calories from saturated fat. Zivkovic et al  [25] suggested increasing the intake of monounsaturated fatty acids (MUFA) instead of SFA (saturated fatty acids) and increasing the intake of complex carbohydrates instead of simple carbohydrates for NAFLD patients. Increasing consumption of n-3 fatty acids, found in fish oil and walnuts, may reduce liver damage in NAFLD patients. Consumption of excessive amounts of sucrose-sweetened soft drinks or fructose [25, 26, 27] may lead to the development of NAFLD. Low-carbohydrate, high-fat ketogenic diets may negatively affect metabolism and cause the development of NASH [22, 26]. Asrih and Jornayvaz [28] suggested the presence of a link between the development of NAFLD to high-fat and ketogenic diets and recommended the consumption of low-refined carbohydrate, low-fat and low-saturated acid-containing diets. Non-digestible and fermentable fibre-containing, low-glycemic index diets may help to improve the health of NASH patients.

Dietary Guidelines for Americans 2020-2025 [29] recommend a daily calorie intake range for healthy women and men of age 19 through 30 to be between 1,800 to 2,400 kcal and 2,400 to 3,000 kcal, respectively. Such a diet should consist of 10-35% protein, 45-65% carbohydrate and 20-35% fat. The amount of fibre in the diet should be adjusted to 14 grams per 1000 calories. The percentage of saturated fat should be kept below 10% of total calories [29]. The Academy of Nutrition and Dietetics [29] recommends 500 to 750 calories/day of deficit for weight loss. The presence of saturated versus unsaturated fat and complex versus simple carbohydrates in a diet is reported to have opposing roles in the development of NAFLD [25, 28]. The insulin sensitivity index and non-alcoholic steatohepatitis increase with the consumption of cholesterol and saturated fat and decrease with the consumption of polyunsaturated fat and fibre [30]. MUFA and polyunsaturated fatty acids (PUFA), particularly n-3 PUFA were reported to reduce liver fat content. In addition, it has been shown that MUFA can reduce body fat accumulation and impair plasma lipid levels, thus may prevent NAFLD development. A diet rich in n-3 PUFA has been found to reduce body weight and accumulation of hepatic triglycerides that causes liver steatosis [31].

Carbohydrates are stored in the body via conversion into fat or glycogen, NAFLD occurs since the fat accumulates in the liver not elsewhere. Excessive fructose consumption in the diet may contribute to the development of NAFLD [25, 28]. Zelber-Sagi et al [32] when evaluating the diet consumed by the NAFLD patients, reported that they had consumed 50% more soft drinks, 27% more meat and less omega-3-rich fish than the control group. Obese rats which consume a low-carbohydrate diet lost more weight than those that consumed a low-fat diet with equal calories, their insulin sensitivity improved and their triglyceride levels decreased, thus their obesity symptoms improved [33]. Low carbohydrate and high-fat consumption by healthy individuals triggered insulin resistance [34]. The higher the damage to the liver, the less should be the conversion efficiency of the glycogen to the energy providing chemicals; therefore, energy conversion efficiency of glycogen should be worse in Child-Pugh Score C patients than that of Child Pugh Score B patients, and that should be worse than Child Pugh Sore A patients.

Weight gain was prevented, but NAFLD and hepatic insulin resistance were observed in mice fed with a ketogenic diet [26]. Low-carbohydrate, high-fat ketogenic diets, besides being effective in weight loss may negatively affect metabolism and have a role in the development of NASH [22].

  • Entropic age

Organisms live at far-from-equilibrium with their surroundings while maintaining homeostasis, importing energy and exergy and exporting entropy, a fraction of the generated entropy accumulates in the body and causes ageing [35]. Inactivation and malfunctioning of the biomolecules are among the consequences of ageing [36]. When organisms accumulate the maximum tolerable entropy, they die and then attain equilibrium with their environment [37]. Humans generate energy for their life processes through the consumption of food [38]. The catabolism of macronutrients, e.g., carbohydrates, fats, and proteins, into their small molecules leads to entropy generation [39]. Öngel et al. [9] calculated the nutrition and disease-related entropy generation based on patient-specific diets and calculated the expected lifespan of the patient groups with different types of cancer. Silva and Annamalai [40] extended this concept and calculated entropy generation and its stress on different organs in the body. Then, Kuddusi [41] calculated the metabolic entropy generation rate as 0.46x10-5 kW/kg K and the expected lifespan of 79 years in the Marmara district of Turkiye. The present study aims to evaluate the effects of intermittent fasting and fatty liver treatment diets on the longevity of patients.

  1. MATERIALS AND METHODS

A schematic description of the thermodynamic system boundaries uptakes and the metabolic waste of the dieting individual is presented in Figure 1.

  • Intermittent fasting diet plans

In the present study, calculations are made for 25 and 50 years old individuals with 170 cm of height and 80 kg of initial weight. Their basal metabolic rates (BMR) were estimated with the Harris-Benedict [42] equation (Table 1) as:

                                                                        (1)

where W represent the weight (kg), H is the height (cm), and A is the age of the subject. The daily caloric need of an individual is calculated as BMR x 1.2. With this approach caloric needs of 25-year-old and 50-year-old individuals were estimated as 2,200 and 2,000 kcal/day, respectively. Diet compositions suggested for these individuals are presented in Table 2. Amounts of the oxygen uptake and the metabolic waste for weight-maintaining, weight gain and weight loss IF diets for the 25 and 50 years old 80 kg individuals are presented in Table 3. When these individuals start intermittent fasting to lose weight and get into a healthy body mass index (BMI) range. BMI is calculated by dividing the body mass of a person by the square of the body height and expressed in units of kg/m². A person is classified as “overweight” with an initial BMI of (80/(1.702)) =28 kg/m2. This individual needs to decrease the BMI to 23 kg/m2 to get into the healthy weight range [43] and when attains the target BMI body weight becomes 66 kg and then the BMR of the 25 years old and the 50 years old subjects become 1,680 and 1,537 W, respectively. One of the most common methods of IF is following the 16:8 plan. In this method, the individual eats for 8 hours a day without any limitation, then do not consume any food, except tea, black coffee and water for 16 hours [16]. Equations (reactions) 2-4 indicate that O2 consumption must increase with the rate of metabolism of the nutrients. Table 3 indicates that O2 consumption increases in case of the weight gain.

2.2. Fatty liver disease diet plans

A schematic drawing thermodynamic system, describing the diet-induced fat accumulation or depletion in the liver is presented in Figure 2. In hepatic diseases, the risk of complications and mortality increase with malnutrition [44]; therefore, preventing malnutrition should be one of the main concerns. Even though NASH is commonly associated with obesity, it does not rule out the risk of malnutrition, furthermore, a sudden decrease in muscle mass in obese patients may be caused by sarcopenic obesity [45]. For obese patients, 5-10% weight loss is recommended, but protein intake must be monitored to prevent muscle mass loss [46]. To avoid hepatocyte necrosis (death of liver cells) and fibrosis from worsening, weight loss is recommended to be less than 1.6 kg per week [47]. Fibrosis may refer to the connective tissue deposition that occurs as part of normal healing or to the excess tissue deposition that occurs as a pathological process. An energy intake of 35-40 kcal/kg per day and a protein intake of 1.2-1.5 g/kg per day is recommended for NAFLD patients [48]. Including late-night snacks and breakfast in the diet might be beneficial in meeting the daily caloric needs [46]. In EASL [46] and ESPEN [48] guidelines, the importance of branched-chain amino acid (BCAA) consumption is highlighted; therefore, the majority of the protein intake should come from dairy protein sources. If the patient cannot uptake enough calories, leucine-enriched BCAA supplements might be used. The consumption of sugared beverages and desserts is strongly discouraged, in addition, sodium intake should be limited as well [49]. In the nutrition of liver patients, zinc and vitamin E supplementation were found highly beneficial [50]. In the present study, diets were planned for 100 kg obese patients (BMI >30) with NAFLD. The caloric need of the patient with NAFLD with Child-Pugh Scores A, B, and C was assumed to be higher with advanced stage due to stress and catabolism process occurring in the body (Table 5).

2.3. Entropy generation and lifespan estimation

2.3.1. Intermittent fasting

Entropy generation rates for a 25-years overweight healthy person and a 50-years overweight healthy individual before and after the IF diet were calculated by using the thermodynamic properties presented in Table 4 by following the same procedure as Öngel et al. [7]. The model was designed by contemplating that two different age groups 80 kg subjects consume the diets causing them to gain weight until their 25 and 50, then they follow the IF diet to lose weight and after the weight loss, they consume the weight-maintaining IF diet. Based on these considerations, three different entropy generation rates were calculated depending on the prepared diets and the lifespan of these subjects was estimated (Table 5).

It was assumed that a healthy person digests 92% of the proteins, 95% of the lipids, and %99 of the carbohydrates provided in the diet [51]. The amount of these macronutrient intakes with the diets is presented in Table 6. In equations (2-4), the metabolism of carbohydrates, lipids and proteins was modelled in terms of glucose (C6H12O6), palmitic acid (C6H32O2), and an average of 20 amino acids (C4.57H9.03N1.27O2.25S0.046), respectively:

                                                C6H12O6 + 6O2 →6H2O + 6CO2                                                               (2)

                                                 C6H32O2 + 23O2 →16CO2 +16H2O                                              (3)

             C4.57H9.03N1.27O2.25S0.046 + 4.75O2 →3.245H2O + 3.935CO2 + 0.635CH4N2O             (4)

Urine and faeces are excreted after metabolism and absorption of nutrients. Based on the same procedure as Öngel et al. [9], the amount of urine was calculated to maintain the urea content as 20 g/L and excrete 65% of the urea out of the body. The amounts of the inhaled O2 and exhaled CO2, and H2O are shown in Table 7. Evaporative loss of H2O via perspiration is not considered in the calculations, since water uptake does not contribute to entropy generation. It was assumed that the patient would drink more water in case of excessive perspiration and it will not affect the results. 

Based on the assumption that nutrients enter the body at 25oC, and H2O and CO2 are excreted from the body at 37 oC, the amount of heat generation rate was found by using equation (6).

                                                                                         (6)

Where,  is the heat released by the body,  refers to the mole number rates of the products exit from the system and  is the mole number rate of the reactants entered the system,  and  represent the enthalpy of formation of the products at the body temperature and at the standard conditions, respectively. According to Silva and Annamalai [68], the metabolism of one mole of each of glucose, palmitic acid, and the average 20 amino acids produce 32, 106 and 8 moles of ATP, respectively. The amount of heat released by their metabolism was found with equation (7); metabolic efficiency (h) of glucose, palmitic acid, and the average of 20 amino acids were taken as 34.6%, 32.2% and 10.4%, respectively [40].

                                                                                                             (7)

Where  is the enthalpy of metabolism. The entropy generation rate of the overweight 25 years and 50 years old subjects were computed by considering their diets via equation (8) and given in Table 5.

                                                             (8)

where, n is the mol number of chemicals taken in or out of the system, s represents the specific entropy of the chemicals, T describes the human body temperature (37oC) and Q defines the heat generated by the body.

Silva and Annamalai [40], found that individuals generate 11,404 kJ/kg K during their whole life. Based on this principle, entropy generation rates at the ages of 25 and 50 were calculated and then their remaining entropy generation rates were computed by adding entropy generation rates. Hwangbo et al. [52], suggested that human lifespan estimations may have an error of 10%-25% depending on genetic factors, environmental factors and the interactions between them. After weight loss IF, it was assumed that longevities of the 25 years old and 50 years old subjects increase between 10%-25% owing to autophagy. Hence, it was assumed that annual entropy generation rates decreased in parallel to these values during the weight-maintaining diet (Table 5).

To perform calculations of the entropy generation rates during weight loss IF diet and weight maintaining intermittent fasting diet, weight of 25 and 50 years old individuals were determined with the help of the “Weight Loss Predictor Calculator” of  Pennington Biomedical Research Center as 70.97 and 71.21 kg for each diet, respectively. Weight versus the time plots is presented in Figure 3.

2.3.2. Fatty liver disease

In the present study, the oxidation reaction of glucose, palmitic acid, and an average of 20 amino acids was chosen to represent the metabolism of carbohydrates, lipids, and protein, respectively and calculations were performed based on reactions Eqs.2-4. All the calculations for NAFLD patients with different Child-Pugh Scores and healthy people were done by using the data given in Table 4 based on the planned diet plans as shown in Table 5. According to the procedure described by Öngel et al. [9], the entropy generation rate and lifespan estimation of the patients were calculated. It is assumed that oxidation of protein, fat, and carbohydrate is %28, %46, and %88 for the patient in Child-Pugh Score A, %31, %55, and %86 for the patients in Child-Pugh Score B and %24, %59, and %88 for the patient with Child-Pugh Score C, respectively [53]. However, it was considered that during the disease, the patients in Child-Pugh Scores A, B, and C lost 5 kg, 10 kg, and 20 kg weight, respectively due to complications. For a healthy person, these rates are %92, %95, and %99 for protein, fat, and carbohydrate, respectively [51]. Based on the entropy balance equation (Eq. 8) by Özilgen and Sorgüven [54] and Kuddusi [41], the entropy generation rate and lifespan estimation were found. It was assumed that    equals zero because the thermodynamic system, the body, is in a quasi-steady-state condition. To calculate the entropy generation rate during NAFLD (Scirrhotic), it was assumed that patients maintained healthy life until their 40 and consumed the diet of a healthy person and then, they became NAFLD patients; therefore, they started consuming a special diet for each Child-Pugh Score (Table 5). The lifespan entropy generation limit was 11,404 kJ/K kg in the calculations of Kuddusi [41]. The one-year survival rate is around %95, %80, and %44 for the patients with Child-Pugh Scores of A, B, and C, respectively [54]. The average lifetime (tavg) was found by considering the one-year survival rate for each Child-Pugh Score was calculated (Table 7).

A similar model to Öngel et al [9] will be employed to calculate the disease-related entropy generation rates of the NAFLD patients at Child-Pugh Scores A, B and C. If that person should be diagnosed with NAFLD at the age of 40, he would have already generated 4,796 kJ/kg K entropy and may generate 6,604 kJ/kg K of more entropy during the rest of his lifespan. Pinter et al [54] indicate that 95%, 80% and 44% of the NAFLD patients with Child-Pugh Score A, B and C will survive the first year of the disease; therefore, when compared to the 119.9 kJ/kg K year of entropy generation rate of an obese (otherwise healthy) person, we may estimate that on the average a Child-Pugh Score A patient may generate (119.9 kJ/kg K) [1/(0.95)]= 126.2 kJ/kg K of entropy annually. Similarly, Child-Pugh Score B and C patients may generate 149.9 and 272.5 kJ/kg K of entropy annually, respectively (Table 8). These numbers were the entropy change of the the metabolism reactions of the nutrients

  1. RESULTS AND DISCUSSIONS

3.1. Intermittent fasting diet

Weight maintaining IF (16:8) diet reduced annual entropy generation rate to 28.3 kJ/kg K and 12.6 kJ/kg K for a 25-year- old individual and a 50-year-old individual in comparison with diet causing weight gain, respectively. Therefore, the longevities of the 25 years old and 50 years old individuals are extended to 33 years and 14 years, respectively (Table 5). Total entropy generation until the age of 25 or 50 and the estimated lifespan of the subjects if they continue consuming the weight loss IF diet until the end of their lifespan are presented in Table 5 with the prevailing BMRs are presented in Table 5. As the age increases the diet and the BMR decreases due to the decrease in calories in the diets.

Goodrick et al. [55], found that the IF diet led to an 83% extension of rat lifespan and the ageing rate and mortality rate were decreased by the IF diet when compared with ad libitum. Ulu et al. [56], concluded that the human lifespan may increase by approximately 3% with the consumption of the IF diet for 30 days based on thermodynamic calculations. In the literature, there is no study about the effects of a long-term IF diet on human lifespan because of this we could not compare our results with the experimental data showing how a long-term IF diet affects human lifespan extension. However, Yang et al. [57] found that the CR diet leads to a reduction of inflammation and upgrades metabolic homeostasis in humans and helps protect the body from the effects of ageing and also slows the rate of the ageing process. Accumulation of certain proteins in the gut, muscle, and neuron cells accelerates ageing; therefore, autophagy appears as an important process and an extension of the lifespan and slows down the ageing process [58]. In a thermodynamic view, dysregulation of autophagy may increase the entropy generation rate [59].

In the present study, approximately 25% and 13.5% of calories were reduced in the weight loss and weight maintaining IF diets, in comparison with the weight gain diets for a 25-year-old individual. The proposed diet reduces these calories by approximately 25% and 15% for a 50-year-old individual. Our results show that the increase in the lifespan of the 25 years old individual depending on IF diet was 19 years higher than that of the 50 years old individual. Ravussin et al. [60], suggested that if an individual starts a 20% CR diet, implying reduction of the calories by 20 %, at 25 years of age and maintains it until 80, the lifespan of that person may increases by approximately 5 years. On the other hand, it extends only two months if an individual follows a 30% CR diet at 55 age and sustains it for 22 years. If the Rhesus monkeys start consuming an early CR diet, they may live longer than the old age onset [61]. The results of our calculations are compatible with those of the previous reports. Willcox et al. [62], argued that the Okinawan diet style increases their 65 years of lifespan by 6% (1.3 years) to 20% (3.6 years) when compared to Japanese and American lifespans, respectively. Okinawans' survival rate did not show an increase only due to CR, but it also depended on diet composition. Yildiz et al. [63], found that more calorie intake changed gut microbiota composition negatively, hence, it caused more entropy generation when compared to the same composition of low-calorie diet consumption. Moreover, Öngel et al. [7], determined that various diet types affected the human lifespan leading to the generation of entropy in different amounts and the Mediterranean diet increased the human lifespan causing a lower level of entropy generation. More protein and fat intake causes a much more entropy generation rate and therefore, lifespan expectancy decreases [9, 40].

During the IF diet period, the oxygen consumption rate of individuals decreases when compared to the diet causing weight gaining (Table 3). It was also found that CR diminished the level of oxidative damage and failure of function related to oxidative damage. Hershey and Lee [37], argued that entropy generation increases in parallel with the basal metabolic rate (BMR), slows with age and prolonged overfeeding cause the increase in BMR and hence the sgen. Our results show that the increase in BMR and entropy generation rate parallels and BMR decreases with weight loss after the IF diets (Table 1). Reduced oxidation energy implies reduced entropy generation.

3.2. NAFLD diets

Çatak et al [64] by using the lifespan entropy generation method estimated the longevity of an obese person who uptakes 10 % more nutrients than a normal person is about 5 years less than that of a non-obese person. The diet lists were prepared for a healthy obese person to consume before she/he becomes NAFLD and for the patients with each Child-Pugh Score of NAFLD. Based on these diets, we calculated patients’ entropy generation rate and lifespan during their healthy life and NAFLD by following the procedure of Öngel et al. [7]. Thermodynamic properties of the macronutrients and their oxidation products which are employed in these calculations are presented in Table 4. According to Table 8, a healthy obese person generates annually 119.9 kJ/kg K per year and 4,796 kJ/kg K of entropy during their 40 years of life. If this person maintains a healthy life and consumes the same diet may live for 94 years. The risk of steatosis development increases depending on BMI. The incidence of steatosis is 65% in obese people with Child-Pugh Scores of A and B [65]. Here, we analyzed the effects of diets prepared for each Child-Pugh Score patient's entropy generation rate and lifespan. Our results showed that a Child-Pugh Score A NAFLD patient generates 126.2 kJ/kg K of entropy and will have 92 years of lifespan. Therefore, it may be assumed that NAFLD with Child-Pugh Score A shortened the lifetime of an obese individual by 2 years. As a result of the calculations, it was found that the NAFLD patients in Child-Pugh Score B may live 84 years by generating annually 149.9 kJ/kg K entropy (Table 7). It can be estimated that the longevity of the obese patient in Child-Pugh Score B can decrease by 10 years. We found that a patient with Child-Pugh Score C may generate 272.5 kJ/kg K year of entropy and have the shortest lifespan (64 years) and his/her life expectancy will be 30 years less when compared with the lifespan estimation of a healthy obese person.

Child-Pugh Score A, B and C patients have good, mid, and poor nutritional status, respectively. In the patient with Child-Pugh Score C, ascites and encephalopathy develop and they are at advanced and refractory levels. They are absent in a patient with Child-Pugh Score A and minimal levels in a patient with Child-Pugh Score B [66]. Our results showed that the patient's lifespan with Child-Pugh Score C shortens by 26 and 15 years compared to the patients with Child-Pugh Scores A and B, respectively.

Schneeweiss et al. [67] suggested that this value and the basal metabolic rate increased in parallel with Child-Pugh Scores, e.g., the oxygen consumption rate of the whole body of the NAFLD patients was (228.8 ± 7.1 ml/min)/1.73 m3 although it was (206.5 ± 4.0 ml)/min/1.73 m3 in normal individuals. The abnormal increase in oxygen consumption causes much more entropy generation. Therefore, the disorder of the liver and finally the whole body increases and ageing occurs and lifespan shortens. Our results also show that oxygen consumption increases with the stage of the NAFLD, and it was the highest in the patient at Child-Pugh Score C (Table 7).

This study is based on the concepts originally suggested by Silva and Annamalai [68, 69], and Annamalai and Silva [40]. Recently Annamalai [70] argued the similarity between oxygen-deficient combustion and metabolism and explained why proteins have low metabolic efficiency (η=10 %) and cause higher entropy generation when compared to carbohydrates and fats (almost η=40). Proteins are used mainly for bodybuilding or replacing cells. In the present study, we have extended this concept to diet treatment with IF and treatment of the liver diseases. Our results pertinent to IF diet are agrees with the findings of Siclair [71], who argue that CR is not simply a passive effect but an active, highly conserved stress response that evolved early in life's history to increase an organism's chance of surviving adversity.

  1. CONCLUSION

In the present study, diet lists were prepared for a healthy obese person and Child-Pugh Score of A, B and C for patients, who got NAFLD after being 40 years old. It was found that healthy obese people generate 119.9 kJ/kg K per year of entropy and in total 4,796 kJ/kg K entropy in their 40 years of life. If a healthy obese person continues to consume the same diet after the age of 40, this person may have 94 years of life expectancy. Meanwhile, wholebody of the Child-Pugh Score A, B and C NAFLD patients may generate 126.2, 149.9 and 272.5 kJ/kg K year of entropy and they would have 92, 84 and 64 years of life expectancy respectively. These results imply that if the patients should continue to consume their present diets, the NAFLD may decrease the lifespan expectancy of  Child-Pugh Score A, B and C patients by 2, 10 and 30 years, respectively. However, if they should make a major shift as recommended in the present study, the lifespan expectancy of  Child-Pugh Score A, B and C patients may increase by 29, 32 and 43 years, respectively. This study was carried out to demonstrate the effects of diets on the lifespan expectancy of NAFLD patients. It should be emphasized that the predictions given here are based on the assumption that the subjects will not develop any other diseases and health complications other than the NAFLD in their lifespans. In the case of such other complications, we would not expect these results to be valid.

NOMENCLATURE

: Enthalpy of formation of the products at the body temperature (J/mole)

: enthalpy of formation of the nutrients at the standard conditions (J/mole)

  = Enthalpy of heat generation rate due to heat released from the metabolic reaction of a nutrient  (J/mole)

: Mole number rates of the products exiting from the system (moles/s)

 is the mole number of the reactants entering the system (moles/s)

: The rate of heat release from the body (J/s)

: Entropy of the nutrients while being uptaken by a person (J/mole K)

: Entropy of the nutrients while being uptaken by a person (J/mole K)

h metabolic efficiency of the nutrients

GLOSSARY OF MEDICAL AND BIOLOGICAL TERMS

ATP: Adenosine triphosphate, an energy-carrying molecule found in the cells of organisms.

Autopagy: A cellular maintenance process that causes the degradation of the damaged components such as misfolded proteins or damaged organelles

BCAA: Branched-chain amino acid

BMR: Basal metabolic rate (the number of calories that the body needs to accomplish its most basic, life-sustaining functions)

Child-Pough Scoring System: Also known as the Child-Pugh-Turcotte score and predicts mortality in cirrhosis patients to guide the selection of patients who would benefit from surgery.  A - good hepatic function, B - moderately impaired hepatic function, and C - advanced hepatic dysfunction.

Cirrhosis: Apearance of the scars in the liver usually in the form of fibrous connective tissue caused by long-term liver damage

CR: Calorie restriction, means proviting less calories to a subject

EEP: Excess entropy production

IF: Intermittent fasting, skipping meals without increasing the calories of the others

MUFA: Monounsaturated fatty acids

NAFLD: Non-alcoholic fatty liver disease (there is fat but no damage in the liver)

NASH: Non-alcoholic steatohepatitis (inflammation of the liver with concurrent fat accumulation)

SFA: Saturated fatty acids

Steatosis: A largely harmless build-up of fat in the liver cells

Ubiquitinatation: An enzymatic process that involves the bonding of an ubiquitin protein to a substrate protein

CONFLICT OF INTEREST STATEMENT: Authors declare no conflict of interest with any parties

ETHICS COMMITTEE STATEMENT: All the data are generated theoretically; therefore, not applicable

  1. REFERENCES

[1] Yildiz C, Bilgin VA, Yilmaz B, Özilgen M. Organisms live at far-from-equilibrium with their surroundings while maintaining homeostasis, importing exergy and exporting entropy. Int J Exergy 2020a;31(3): 287-300; doi:10.1504/ijex.2020.10027921.

 [2] Aoki I, Entropy production in human life span: A thermodynamical measure for aging. Age 17, 29-31, 1994

 [3] Caliskan, H. Energetic and exergetic comparison of the human body for the summer season. Energy Convers. 76, 169-176, 2013; doi.org/10.1016/j.enconman.2013.07.045.

 [4] Mady CEK, Ferreira MS, Yanagihara JI, de Oliveira S. Human body exergy analysis and the assessment of thermal comfort conditions. International Journal of Heat and Mass Transfer 2014; 77: 577–584.

 [5] Molliet, DS, Mady CEK. Exergy analysis of the human body to assess thermal comfort conditions: comparison of the thermal responses of males and females, Case Stud. Therm. Eng.; 25(100912), 1-8, 2021; doi:org/10.1016/j.csite.2021.100972.

[6] Kayalı D, Yavuz Y, Yilmaz B, Özilgen M. Entropic assessment of sleeping comfort. International Journal of Thermodynamics. 2022; 25(3), 64-73; doi:10.5541/ijot8911.

 [7] Öngel ME, Yildiz C, Akpınaroğlu C, Yılmaz B, Özilgen M. Why women live longer than men do? A telomere-length regulated and diet - based entropic assessment. Clin Nutr, 2020; S0261-5614(20): 30395-2; doi:10.1016/j.clnu.2020.07.030.

 [8] Yildiz C, Öngel ME, Bayram Yilmaz B, Özilgen M. Diet-dependent entropic assessment of athletes’ lifespan. J Nutr Sci; 2021;10(10), E83; doi:10.1017/jns.2021.78.

[9] Öngel M.E, Yildiz C, Yilmaz C, Özilgen M. Nutrition and disease-related entropy generation in cancer. Int J Exergy, 2021; 34(4), 411-423; doi:10.1504/IJEX.2021.114091.

 [10] Ulu G, Öngel M.E, Yilmaz B, Özilgen M. Thermodynamic evaluation of pregnancy and lactation. International Journal of Thermodynamics, 2022; 25(4):45-54; doi:10.5541/ijot.1145655.

[11] Yildiz C, Özilgen M, Why brain functions may deteriorate with aging: A thermodynamic evaluation Int J Exergy, 2022;37(1); doi:10.1504/IJEX.2022.120.109.

[12] Cotter TG, Rinella M. Nonalcoholic fatty liver disease 2020: The state of the disease. Gastroenterology, 2020;158(7), 1851-1864; doi:org/10.1053/j.gastro.2020.01.052.

[13] Cirrhosis Collaborators. The global, regional, and national burden of cirrhosis by cause in 195 countries and territories, 1990-2017: a systematic analysis for the global burden of disease study 2017. Lancet Gastroenterol Hepatol, 2020;5(3):245-266; doi:10.1016/S2468-1253(19)30349-8.

[14] Beard DA, Qian H 2005. Thermodynamic-based computational profiling of cellular regulatory control in hepatocyte metabolism. Am J Physiol Endocrinol Metab, 288:E633–E644; doi:10.1152/ajpendo.00239.2004

[15] Kim BH, Joo Y, Kim M-S et al. Effects of intermittent fasting on the circulating levels and circadian rhythms of hormones. Endocrinol Metab., 2021; 36:745-756; doi:org/10.3803/EnM.2021.405

 [16] Patterson RE, Sears D. Metabolic effects of intermittent fasting. Annu Rev Nutr,   2017;37:371-393; doi:10.1146/annurev-nutr-071816-064634.

[17] Fisher FM, Maratos-Flier E. Understanding the physiology of FGF21. Annu Rev Physiology,   2016;78:223-41; doi:10.1146/annurev-physiol-021115-105339.

[18] Cabo R, Mattson M.Effects of intermittent fasting on health, aging, and disease. New  England Journal of Medicine, 2019;381:2541-51; doi: 10.1056/NEJMra1905136.

[19] Kitada M, Koya D. Autophagy in metabolic disease and ageing. Nat Rev Endocrinol, 2010; 17:647–661; doi:org/10.1038/s41574-021-00551-9.

[20] Cortez-Pinto H, Jesus L, Barros H, Lopes C, Moura MC, Camilo ME. How different is the dietary pattern in non-alcoholic steatohepatitis patients? Clin Nutr, 2006;25(5):816-823; doi:org/10.1016/j.clnu.2006.01.027.

 [21] Eslam M, Newsome PN, Sarin SK. A new definition for metabolic dysfunction- associated fatty liver disease: An international expert consensus statement. J Hepatol, 2020; 73: 202-209; doi:org/10.1016/j.jhep.2020.03.039.

 [22] Kalaitzakis E, Bosaeus I, Ohman L,  Björnsson E. Altered postprandial glucose, insulin, leptin, and ghrelin in liver cirrhosis: Correlations with energy intake and resting energy expenditure. Am J Clin Nutr, 2007; 85, 808-15; doi:org/10.1093/ajcn/85.3.808.

[23] Tsoris A, Marlar CA. Use of the Child Pugh score in liver disease. StatPearls Publishing, Treasure Island, FL, 2021.

[24] AHA (2021), Saturated fat, available at https://www.heart.org/en/healthy-living/healthy-eating/eat-smart/fats/saturated-fats; accessed September 8, 2021.

[25] Zivkovic AM, German JB, Sanyal AJ. Comparative review of diets for the metabolic syndrome: implications for nonalcoholic fatty liver disease,  Am J Clin Nutr, 2007;86 (2):285- 300; doi:org/10.1093/ajcn/86.2.285.

[26] Jornayvaz FR, Jurczak MJ, Lee H-Y, Birkenfeld AL, Frederick DW, Zhang D, Zhang X-M, Smuel V-T, Shulman GI. A high-fat, ketogenic diet causes hepatic insulin resistance in mice, despite increasing energy expenditure and preventing weight gain. Am J Physiol Endocrinol Metab, 2010;299(5):E808-E815; doi:org/10.1152/ajpendo.00361.2010.

[27] Ouyang X, Cirillo P, Sautin Y, McCall S, Bruchette JL, Diehl AM, Johnson RJ, Abdelmalek MF. Fructose consumption as a risk factor for non-alcoholic fatty liver disease. J Hepatol. 2008;48(6):993-999; doi:10.1016/j.jhep.2008.02.011

[28] Asrih M, Jornayvaz FR. Diets and nonalcoholic fatty liver disease: the good and the bad. Clin Nutr. 2014;33(2):186-190; doi:10.1016/j.clnu.2013.

[29] USDA (2020). Dietary Guidelines for Americans 2020-2025, Available at https://www.dietaryguidelines.gov/sites/default/files/2021-03/Dietary_Guidelines_for_Americans-2020-2025.pdf, accessed September 9, 2021.

[30] Musso G, Gambino R, De Michieli F, Cassader M, Rizzetto M, Durazzo M, Faga E, Silli B, Pagano G. Dietary habits and their relations to insulin resistance and postprandial lipemia in nonalcoholic steatohepatitis. Hepatology, 2003;37(4):909-916; doi:10.1053/jhep.2003.50132. 

[31] Levy JR, Clore JN, Stevens W. Dietary n-3 polyunsaturated fatty acids decrease hepatic triglycerides in Fischer 344 rats. Hepatology 2004;39(3):608-616; doi:org/10.1002/hep.20093.

[32] Zelber-Sagi S, Nitzan-Kaluski D, Goldsmith R, Webb M, Blendis L, Halpern Z, et al. Long term nutritional intake and the risk for non-alcoholic fatty liver disease (NAFLD): a population based study. J Hepatol, 2007;47(5): 711-7; doi:org/10.1016/j.jhep.2007.06.020.

[33] Al-Obaidi ZAF, Erdogan CS, Sümer E, Özgün HB, Gemici B, Sandal S, Yilmaz B. Investigation of obesogenic effects of hexachlorobenzene, DDT and DDE in male rats. Gen Comp Endocrinol. 2022;1;327:114098; doi:10.1016/j.ygcen.2022.114098.

[34] Bisschop PH, de Metz J, Ackermans MT, Endert E, Pijl H, Kuipers F, et al. Dietary fat content alters insulin-mediated glucose metabolism in healthy men. Ame J Clin Nutr 2001;73(3):554-9; doi:org/10.1093/ajcn/73.3.554.

[35] Yildiz C, Semerciöz AS, Yalçinkaya BH, Ipek, TD, Öztürk-Özışık, E, Özilgen M. Entropy generation and accumulation in biological systems. Int J Exergy 2020b;33(4):444-468; doi:10.1504/IJEX.2020.111691.

[36] Hayflick L.  Biological aging is no longer an unsolved problem. Ann NY Acad Sci, 2007a; 1100 (1): 1–13; doi:10.1196/annals.1395.001.

[37] Hershey D, Lee WE.  Entropy, aging and death. Syst Res, 1987;4(4), 269-281; doi:org/10.1002/sres.3850040406.

[38] Semerciöz AS, Yılmaz B, Özilgen M. Thermodynamic assessment of the allocation of the energy and exergy of the nutrients for the life processes during pregnancy. Brit J Nutr 2020; 124(7):742-753; doi:org/10.1017/S0007114520001646.

[39] Fine EJ. Feinman RD. Thermodynamics of weight loss diets. Nutrition and Metabolism 1 (2004)15; doi:10.1.1186/1743-7075-1-15

[40] Annamalai K, Silva CA. Entropy stress and scaling of vital organs over life span based on allometric laws. Entropy 14(12): 2550e77, 2012; doi:org/10.3390/e14122550.

[41] Kuddusi L. Thermodynamics and lifespan estimation. Energy, 2015;80, 227-238; doi:org/10.1016/j.energy.2014.11.065.

[42] Harris A, Benedict F. A biometric study of human basal metabolism. PNAS, 1918; 4(12): 370-373; doi;org/10.1073/pnas.4.12.370.

 [43] NIH. Expert panel on the identification, evaluation, and treatment of overweight and obesity in adults clinical guidelines on the identification, evaluation, and treatment of overweight and obesity in adults—the evidence report. NIH Obes Res, 1998; 6(12):51S-209S.

 [44] Teiusanu A, Andrei M, Arbanas T, Nicolaie T, Diculescu M. Nutritional status in cirrhotic patients. Maedica (Bucur). 2012;7(4):284-289; doi:10.1504/IJEX.2020.111691.

[45] Montano-Loza AJ, Angulo P, Meza-Junco J. et al. Sarcopenic obesity and myosteatosis are associated with higher mortality in patients with cirrhosis. J Cachexia Sarcopenia Muscle, 2016; 7(2), 126-135; doi:org/10.1002/jcsm.12039.

[46] EASL. European Association for the Study of the Liver. EASL clinical practice guidelines on nutrition in chronic liver disease.  J Hepatol, 2019;70(1), 172–193; doi:org/10.1016/j.jhep.2018.06.024.

[47] Andersen T, Gluud C, Franzmann MB, Christoffersen P. Hepatic effects of dietary weight loss in morbidly obese subjects. J Hepatol 1991;12(2): 224-229; doi:10.1016/0168-8278(91)90942-5.

 [48] Plauth M, Cabre E, Riggio O et al. (2006). ESPEN Guidelines on enteral nutrition: Liver disease. Clin Nutr, 2006;25(2), 285-294; doi:org/10.1016/j.clnu.2006.01.018.

[49] McClain CJ. Nutrition in patients with Cirrhosis. Gastroenterol Hepatol, (N Y). 2016; 12(8): 507–510.

[50] Hanje AJ, Fortune B, Song M, Hill D, McClain C. The use of selected nutrition supplements and complementary and alternative medicine in liver disease. Nutr Clin Pract, 2006;21(3):255-272. doi:10.1177/0115426506021003255.

[51] Feher J. Energy balance and regulation. In. Feher, J., Quantitative human physiology, 1st ed., page 744-756, Academic Press, 2012.

[52] Hwangbo D-S, Lee H-Y, Abozald LS, Min K-J. Mechanisms of lifespan regulation by calorie restriction and intermittent fasting in model organisms. Nutrients, 2020; 12(4):1194-1218; doi:10.3390/nu12041194.

[53] Davidson HM, Richardson R, Sutherland D, Gordon OJ. Macronutrient preference, dietary intake, and substrate oxidation among stable cirrhotic patients. J Hepatol, 1999;29(5): 1380-1386;  doi:org/10.1002/hep.510290531.

[54] Özilgen, M., Sorgüven, E. (2016). Biothermodynamics. CRC Press Inc. Florida, USA

 [55] Goodrick CL, Ingram DK, Reynolds MA, Freeman JR, Cider CL. Effects of intermitting fasting upon growth and life span in rats. Gerontology, 1982; 28: 233-241; doi:10.1159/000212538.

[56] Ulu G, Semerciöz AS, Özilgen M. Energy storage and reuse in biological systems: Case studies. Energy Storage, 2021; e253; doi:10.1002/est2.253.

[57] Yang L, Licastro D, Cava E, Veronese N, Spelta F, Rizza W, Bertozzi B, Villareal DT, Hotamisligil G, Holloszy JO, Fontana L. Long-term calorie restriction enhances cellular quality-control processes in human skeletal muscle. Cell Reports, 2016; 14(3):422-428; doi:org/10.1016/j.celrep.2015.12.042.

 [58] Koyuncu S, Loureiro R, Lee HJ et al. Rewiring of the ubiquitinated proteome determines ageing in C.elegans. Nature, 2021; 596(7871):285-290; doi:10.1038/s41586-021-03781-z. 

[59] Fiddan-Green RG. Thermodynamic considerations in management of breast cancer. bmj.com, 9 May 2009; eLetter re: Morrow M. Minimally invasive surgery for breast cancer. BMJ, 2009; 338:6557; doi:org/10.1136/bmj.b557.

 [60 E, Gilmore LA, Redman LM. Calorie restriction in humans: Impact on human health. In Ed. Malavolta M, Mocchegiani  M. Molecular basis of nutrition and aging. Academic Press, 2016; 677-692; doi:10.1016/b978-0-12-801816-3.00048-0. 

[61] Mattison JA, Colman RJ, Beasley TM, Allison DB et al. Caloric restriction improves health and survival of rhesus monkey. Nature Communications, 2017; 8:14063; doi:10.1038/ncomms14063.

[62] Willcox BJ, Willcox DC, Todoriki H, Fujiyoshi A, Yano K, He Q, Curb JD, Suzuki M. Caloric restriction, the traditional Okinawan diet, and healthy aging: the diet of the world's longest-lived people and its potential impact on morbidity and life span. Ann N Y Acad Sci. 2007; 14:434-455; doi:10.1196/annals.1396.037. 

[63] Yildiz C, Yilmaz B, Özilgen M. Fraction of the metabolic ageing entropy damage to a host may be flushed out by gut microbiota.  Int J Exergy, 2021;  34(2): 179-195; doi:10.1504/IJEX.2021.113004.

 [64] Çatak J, Develi AÇ, Sorgüven E, Özilgen M,  Inal HS. Lifespan entropy generated by the masseter muscles during chewing: An indicator of the life expectancy? Int J Exergy 2015; 18(1), 46-66; doi:org/10.1504/IJEX.2015.072056.

 [65] Fabbrini E, Sullivan S, Klein S. Obesity and nonalcoholic fatty liver disease: Biochemical, metabolic and clinical implications. J Hepatol, 2010; 51(2), 679–689; doi:org/10.1002/hep.23280.

[66] Durand F, Valla D. Assessment of the prognosis of cirrhosis: Child-Pugh versus MELD. J Hepatol, 2005; 42: S100–S107; doi:org/ 10.1016/j.jhep.2004.11.015.

 [67] Schneeweiss B, Graninger W, Ferenci P, Eichinger S, Grimm G, Schneider B, Laggner AN, Lenz K, Kleinberger G. Energy metabolism in patients with acute and chronic liver disease. Hepatology, 1990;11(3): 387-393; doi:org/10.1002/hep.1840110309.

 [68] Silva CA, Annamalai K. Entropy generation and human aging: Lifespan entropy and effect of physical activity level. Entropy, 2008;10: 100-123; doi:org/10.3390/entropy-e10020100.

[69] Silva CA, Annamalai K. Entropy generation and human aging: lifespan entropy and effect of diet composition and caloric restriction diets, J Thermodyn 2009;2009(186723), 1-10; doi:10.1155/2009/186723.

[70] Annamalai K. Oxygen deficient (OD) combustion and metabolism: Allometric laws of organs and Kleiber’s law from OD metabolism? Systems 2021;9, 54, doi:org/10.3390/systems 9030054.

[71] Sinclair DA. Toward a unified theory of caloric restriction and longevity regulation. Mech Ageing Dev. 2005;126(9):987-1002; doi:10.1016/j.mad.2005.03.019. 

Table 1. Estimated BMR for each person (calculated with the BMR of the calculator net, available at https://www.calculator.net/about-us.html; accessed December 30, 2022.

Subjects

Initial BMR

After IF BMR

25 years old men

1,885 kcal/day

7,879 kJ/day

1,680 kcal/day

7,022 kJ/day

50 years old imen

1,618 kcal/day

6,763 kJ/day

1,478 kcal/day

6,178 kcal/day

Table 2. The estimated composition of each diet

Type of diet

Calorie (kcal)

Carbohydrates (g)

Protein (g)

Lipid (g)

Weight maintaining IF diet for a 25 years old, 80 kg person

2,210

318

99

59

Weight maintaining IF diet for a 50 years old, 80 kg person

2,010

295

86

54

Weight gain IF diet for a 25 years old 80 kg person

1,912

276

94

48

Weight gain IF diet for a 50 years old 80 kg person

1,711

257

74

43

Weight loss IF diet for a 25 years old 80 kg person

1,663

242

68

47

Weight loss IF diet for a 50 years old 80 kg person

1,510

225

67

38

Table 3. Amounts of the oxygen uptake and the metabolic waste for the weight maintaining, weight gain and weight loss IF diets for the 25 and 50 years old 80 kg individuals.

O2

(g/day)

H2O

(g/day)

CO2 (g/day)

Urine

(g/day)

Dry feces (g/day)

Weight maintaining IF diet for a 25 years old 80 kg person

532.9

257.6

652.0

894.7

1646.5

Weight maintaining IF diet for a 50 years old 80 kg person

475.7

232.0

584.6

677.0

1856.1

Weight gain diet for a 25 years old 80 kg person

613.2

296.6

748.4

942.3

1601.8

Weight gain diet for a 50 years old person

560.0

271.7

684.6

786.8

1751.9

Weight loss IF diet for a 25 years 80 kg person

463.7

224.7

565.2

647.2

1883.6

Weight loss IF diet for a 50 years 80 kg person

420.0

204.5

515.7

613.0

19016.9

Table 4. Thermodynamic properties of the macronutrients and the products of their oxidation at 1 atm (adapted from Kuddusi  [69])

Enthalpies (h} and entropies of the macronutrients (s) of O2, CO2, and H2O at 1 atm and 298 K and 310 K

Chemical

(kJ/kmol)

s at (298K)

(kJ/kmol K)

(kJ/kmol)

s at (310K)

(kJ/kmol K)

C6H12O6 (glucose)

-1,260 x 103

212

-

C16H32O2 (palmitic acid)

-835x103

452

-

C4.57H9.03N1.27O2.25S0.046 (average of the 20 amino acids)

-385x103

1.401x119

-

O2

8,682

218

220

H2O

10,302

219

CO2

9,807

243

Table 5. Entropy generation rates and lifespan estimation by 25 and 50 years individuals depending on the diets

25 years old individual

50 years old individual

 Total entropy generation until the age of 25 or 50 and BMR

2,560 kJ/kg K

7,879 kJ/day

4,651 kJ/kg K

6,763 kJ/day

Annual entropy generation due to consumption of the weight gain IF diet (kJ/kg K year)

102.4

135

Annual entropy generation rate due to IF weight maintaining diet (kJ/kg K year)

81.1

121

Annual entropy generation rate due to weight loss IF diet (kJ/kg K year)

74.1

91

Lifespan estimated (years) when they continue consuming the IF weight gain diet until the end of their lifespan

110

100

Lifespan estimated (years) when they continue consuming IF weight maintaining diet until the end of their lifespan

133

106

Estimated Lifespan of the subjects if they continue consuming the weight loss IF diet until the end of their lifespan

143 years

7,022 kJ/day

135 years

6,178 kcal/day

Table 6. Diet plans for obesity-induced NFLD patients based on Child-Pugh Score and healthy obese person

Child-Pugh Scores

Child-Pugh Score A

Child-Pugh Score B

Child-Pugh Score C

Healthy obese person

kcal/day

3,000

3,200

3,300

3,100

Carbohydrate (g/day)

428

416

470

430

Protein (g/day)

130

160

165

150

Fat (g/day)

92

110

79

98

Total (g)

650

686

714

687

Table 7. The amount of the consumed oxygen, exhaled carbon dioxide, metabolic water, and the excreted waste

Healthy obese person

Child-Pugh Score A

Child-Pugh Score B

G Child-Pugh Score C

O2 (g/day)

898

574

619

626

H2O (g/day)

428

294

307

320

CO2 (g/day)

1,081

728

763

793

Dry feces  (g/day)

1,078

2,326

2,221

233

Urine (g/day)

1,428

377

513

394

Table 8.  Entropy generation rates and lifespan estimations for healthy obese and Child-Pugh Score A, B and C people

Healthy obese person

Child-Pugh Score A patient

Child-Pugh Score B patient

Child-Pugh Score C patient

The annual entropy generation rate  until age 40 (kJ/kg K year)

119.9

119.9

119.9

119.9

Total entropy generation in 40 years Shealthy (kJ/kg K)

4,796

4,796

4,796

4,796

The annual entropy generation rate  until after 40 in the case of no diet change (kJ/kg K year)

119.9

126.2

149.9

272.5

The annual entropy generation rate after the age of 40 in the case of a change of the diet (kJ/kg K year)

122

76.4

87.2

98.8

Expected lifespan in case of no diet change (years)

95

92

84

64

Expected lifespan in case of the change of the diet (years)

-

127

116

107

Figure 1. Schematic description of the thermodynamic system boundaries uptakes and the metabolic waste of the dieting individual

  Outgoing blood

                        Incoming Blood

Figure 2. The thermodynamic system describing the fat accumulation in the liver.

Figure 3. Weight of the individuals on the IF diet (------- : 25 years old patiet,  -------: 50 years old patiet)

Reviewer 2 Report

The manuscript entropy-2085457 deals with entropy determinations of dietary plans in different control person groups.
In general, the topic is interesting, but i stopped reading, reasons see below. I cannot further review this manuscript as this manuscript is not properly prepared. Revision is required to review this manuscript in detail.

1) abstract: what is missing is one sentence on the methods used, but rather the authors present a number that is just a x4 multiplier is not worth it.  abstract must be revised.

2) english must be improved. a person is called "HE". why? singular/plural issues must be resolved. what is bodies? (ketone bodies?). i dont understand.

3) Introduction: introducing the 2nd law would be nice to state that methods like thermodynamic flux analyses explain the appearance of pathways, e.g. Scientific reports 11 (1), 1-9. Instead, the authors use >50 !! refs dealing with illness etc, which is just irrelevant. 50% of the refs. should be removed from the introduction.

4) abbreviations are not properly introduced, e.g. BMR, BMI

5) Tables are not explained / cited in the manuscript. This was the point where i stopped reading. Table 2 is not explained at all. what is h(bar) and h(bar)° in table 2? theory on this is missing completely. Please correct for this. Then i can do a follow up review.

Round 2

Reviewer 1 Report

Thermodynamic assessment of the effects of intermittent fasting and fatty liver disease diets on longevity

By Öngela et al , Revised Version

The  revised manuscript  is a vast improvement over the earlier version. The authors addressed most of my concerns, revised paper to provide more clarity, included more refs. and glossary of medical terms and acronyms. I have only a few minor comments

The topic is  interesting and addresses how CR will improve life span even though recommended for NAFLD A, B and C.

Minor Comments:

1.       If Figure is available on weight  gain vs days, please include it. Still Fig 3 does not have weight gain plot

2.       C12 and R12:  I am not sure still why 119.9/0.44 ( survival fraction first year); if 56 % died first year their entropy generation is very high? Did they (NAFLD C) undergo liver transplantation and only 44 % survived?  Clarify

3.       R6: Fatty liver seems to affect conv. Efficiency of glycogen. Any data on conv. Efficiency of glycogen  for NAFLD A, B ,C?  Is conv. Eff lowest for C? if Conv. Eff. Is low ATP production affected?  Clarify

4.       Eq. 3 I is still erroneous. Check C balance: 16 CO2 but only 6 C atoms in?

5.       R23 My comment on Eq.4 is still valid. I meant SO3 not ozone.

6.       You had provided data on Okinawa diet but not Mediterranean Diet: provide more information. style of eating? See https://www.cnn.com/cnn-underscored/health-fitness/mediterranean-diet-food-list

   “Numerous studies have found the Mediterranean diet can reduce the risk for diabetes, high cholesterol, dementia, memory loss, depression and breast cancer”. The diet  is not part of CR but  a healthier composition  with better heart and longer life.

“The diet features … plant-based cooking, with the …focus on fruits and vegetables (% if any) , whole grains, beans and seeds(% if any), , meat (% if any)… extra-virgin olive oil.  (% if any).

7.       Table 7: A, B,C  lost 5, 10, 20 kg. but O2 consumed is 374, 619, 626 ; more O2 more Calories. Why weight loss? Sgen high for C consistent with Table 8

8.       % Reduction in electron transfer= % reduction in O2 consumed see Popovic , Heliyon;  Annamalai, 2021  System

9.       R6 seems to indicate that low CH diet is preferable to low fat diet? Clarify

Author Response

RESPONSE TO THE COMMENTS OF REVIEWER 1 

R1C1) If figure is available on weight gain vs days, please include it. Still Fig 3 does not have weight gain plot

R1R1) We do not have a weight gain plot. Figure 3 is a weight versus time plot prepared for the weight loss IF diet.  

R1C2) C12 and R12:  I am not sure still why 119.9/0.44 (survival fraction first year); if 56 % died first year their entropy generation is very high? Did they (NAFLD C) undergo liver transplantation and only 44 % survived?  Clarify

R1R2) The following lines are added to the manuscript:

The one-year survival rate is approximately 95%, 80%, and 44% for the patients with Child-Pugh Scores of A, B, and C, respectively [54]. Öngel et al [9] estimated the disease-related entropy generation for 19 different varieties of cancer as:

(Disease related annual entropy generation rate by a patient) = (Total entropy generation by the patients in Dt years of remaining life span) / [(fraction of the patients surviving after Dt years) (Dt)].

The same expression is employed in this study for the NAFLD and the remaining average lifetime (tavg) was estimated for each Child-Pugh Score patient and listed in Table 7. If a person is diagnosed with Child-Pough score C NAFLD at the age of 40, he would have already generated 4,796 kJ/kg K entropy and may generate 6,604 kJ/kg K of more entropy during the rest of his lifespan. Pinter et al [54] indicate that 95%, 80% and 44% of the NAFLD patients with Child-Pugh Score A, B and C will survive the first year of the disease; therefore, when compared to the 119.9 kJ/kg K year of entropy generation rate of an obese (otherwise healthy) person, we may estimate that on the average a Child-Pugh Score A patient may generate (119.9 kJ/kg K) [1/(0.95)]= 126.2 kJ/kg K of entropy annually. Similarly, Child-Pugh Score B and C patients may generate 149.9 and 272.5 kJ/kg K of entropy annually, respectively (Table 8).

R1C3) R6: Fatty liver seems to affect conv. Efficiency of glycogen. Any data on conv. Efficiency of glycogen for NAFLD A, B, C?  Is conv. Eff lowest for C? if Conv. Eff. Is low ATP production affected?  Clarify

R1R3) The following lines are added to the manuscript:

Low carbohydrate and high-fat consumption by healthy individuals triggered insulin resistance [34]. The higher the damage to the liver, the less should be the conversion efficiency of the glycogen to the energy providing chemicals; therefore, energy conversion efficiency of glycogen should be worse in Child-Pugh Score C patients than that of Child Pugh Score B patients, and that should be worse than Child Pugh Sore A patients.

R1C4)    Eq. 3 I is still erroneous. Check C balance: 16 CO2 but only 6 C atoms in?

R1R4) Corrected as requested: C6H32O2 + 13O2 →6CO2 +16H2O                                             

R1C5) My comment on Eq.4 is still valid. I meant SO3 not ozone.

R1R5) The following lines are added to the manuscript:

Calculations of this study are based on the metabolic reactions presented in Equations 2, 3 and 4. Carbon and hydrogen appears in all of these reactions and sulfur appears only in equation 4, in the molecular formula of the “average of 20 amino acids”, where the ratio of S to C is, 0.01, and the ratio of S to H is 0.005; indicating that the S compounds to entropy generation are extremely small; therefore, are not considered in the calculations.

R1C6) You had provided data on Okinawa diet but not Mediterranean Diet: provide more information. style of eating? See https://www.cnn.com/cnn-underscored/health-fitness/mediterranean-diet-food-list

R1R6) We have already published a highly detailed analyses on Mediterranean diet as presented in reference [7]:

Öngel ME, Yildiz C, Akpınaroğlu C, Yılmaz B, Özilgen M. Why women live longer than men do? A telomere-length regulated and diet - based entropic assessment. Clin Nutr, 2020; S0261-5614(20): 30395-2; doi:10.1016/j.clnu.2020.07.030.

Repeating the same analyses here would be redundant; therefore, we cited the reference and explained briefly what we have done there.

Öngel et al [7] presented the life expectancy of the women with a telomere-length regulated and diet - based entropic assessment on Mediterranean, Western (American), ketogenic and vegan diets based on the methods presented by Hayflick [36], Annamalai and Silva [40] and Silva and Annamalai [69, 70]. With all of these diets is lifespan estimation of the women were longer than those of the men. Faster shortening of the telomere lengths in men was the major reason of the shorter life expectancy. The highest and the lowest life expectancy for women were estimated with Mediterranean and the vegetarian diets, respectively; men were estimated to have the longest life span with the vegetarian diet and the shortest life span with the ketogenic diet.

R1C7) “Numerous studies have found the Mediterranean diet can reduce the risk for diabetes, high cholesterol, dementia, memory loss, depression and breast cancer”. The diet is not part of CR but a healthier composition with better heart and longer life.

R1C7) The following lines and references are added to the manuscript:

It was also found that caloric restriction diminishes the level of oxidative damage and failure of function related to oxidative damage [64].

[64] Ungvari Z, Parrado-Fernandez C, Csiszar A, de Cabo R. Mechanisms underlying caloric restriction and lifespan regulation: implications for vascular aging. Circ Res. 2008;102(5):519-28; doi:10.1161/CIRCRESAHA.107.168369.

Numerous studies have found the Mediterranean diet can reduce the risk for diabetes, high cholesterol, dementia, memory loss, depression and breast cancer [71]

[71] Roman GC, R.E. Jackson RE, Gadhia R, Roman AN, Reis J. Mediterranean diet: The role of long-chain v-3 fatty acids in fish; polyphenols in fruits, vegetables, cereals, coffee, tea, cacao and wine; probiotics and vitamins in prevention of stroke, age-related cognitive decline, and Alzheimer disease. Rev Neurolo (Paris), 2019; 175: 724 – 741; doi:10.1016/j.neurol.2019.08.005.

R1C8) The diet features … plant-based cooking, with the …focus on fruits and vegetables (% if any), whole grains, beans and seeds (% if any), meat (% if any) extra-virgin olive oil.  (% if any).

R1R8) To our knowledge such results are not available in the literature.     

R1C9) Table 7: A, B, C lost 5, 10, 20 kg. but O2 consumed is 374, 619, 626; more O2 more Calories. Why weight loss? sgen high for C consistent with Table 8.

R1R9) The following paragraph was available in the manuscript; it was revised to respond the comment of the reviewer:

Schneeweiss et al. [68] suggested that this value and the basal metabolic rate increased in parallel with Child-Pugh Scores, e.g., the oxygen consumption rate of the whole body of the NAFLD patients was (228.8 ± 7.1 ml/min)/1.73 m3 although it was (206.5 ± 4.0 ml)/min/1.73 m3 in normal individuals. The abnormal increase in oxygen consumption causes much more entropy generation. Therefore, the disorder of the liver and finally the whole body increases and ageing occurs and lifespan shortens. Our results also show that oxygen consumption increases with the stage of the NAFLD, and it was the highest in the patient at Child-Pugh Score C (Table 7). Higher basal metabolic rate implies higher energy utilization to maintain the live functions, such as heart beat, respiratory, neuronal and renal activity; therefore, the weight of the patients seems like decreasing with oxygen utilization.

R1C10) % Reduction in electron transfer= % reduction in O2 consumed see Popovic, Heliyon; Annamalai, 2021 System

R1R10) The most comprehensive study, regarding this observation was reported by Annamalai [72] and it is the most important reference in the present study:

[72] Annamalai K. Oxygen deficient (OD) combustion and metabolism: Allometric laws of organs and Kleiber’s law from OD metabolism? Systems 2021;9, 54, doi:org/10.3390/systems 9030054.

R1C11) R6 seems to indicate that low CH diet is preferable to low fat diet? Clarify

R1R11) According to reference [33] this statement is correct, if the purpose is weight loss.

Reviewer 2 Report

The revised version improved strongly over the original manuscript. I have some comments left:

# I suggested to cite work that shows how to obtain consistent thermodynamic standard data, e.g. Scientific reports 11
(1), 1
-9.

# revise appendix B, e.g. sin and sout have the same meaning (copy/paste error?), and hf and h is both enthalpy of formation (f is formation in both symbols, but zero denotes standard state).

# as a follow up, this equation is wrong. The authors are encouraged to read https://doi.org/10.1146/annurev-chembioeng-080615-034704. In this paper, the difference between standard state and any other state is explained for Gibbs energy (it can be transferred to enthalpy as well). Reaction enthalpy (eq. 6) mixes up standard state and non-equilibrium states, using one for the products and one for the reactants. This is not consistent. The same treatment must be applied for both, reactants r and products p.

# the format of the figures 1 and 3 is very poor. Increase quality and make a more appropriate figure format, e.g. y-axis: make 5kg steps, no figure header (there is already a figure caption)

Please revise equation 6 and check the results on changes. Then i can do a follow up review.

Author Response

RESPONSE TO THE COMMENTS OF REVIEWER 2

C1R2) The revised version improved strongly over the original manuscript. I have some comments left:

R1R2) Thank you for the appreciation

C2R2) I suggested to cite work that shows how to obtain consistent thermodynamic standard data, e.g. Scientific reports 11 (1), 1-9.

R2R2) This information is not sufficient to cite this open access article, either an article number or doi number is needed. Moreover, our thermodynamic data is not based on measurement of any type, but calculated based on carbohydrate, fat and protein contents of a detailed diet list prepared according to the published guidelines of the nutrition authorities.

C3R2) revise appendix B, e.g. sin and sout have the same meaning (copy/paste error?), and hf and h is both enthalpy of formation (f is formation in both symbols, but zero denotes standard state).

R3R2) In the “Nomenclature”,  is corrected as the “entropy of the excreted molecules (J/mole K)”

C4R2) as a follow up, this equation is wrong. The authors are encouraged to read https://doi.org/10.1146/annurev-chembioeng-080615-034704. In this paper, the difference between standard state and any other state is explained for Gibbs energy (it can be transferred to enthalpy as well). Reaction enthalpy (eq. 6) mixes up standard state and non-equilibrium states, using one for the products and one for the reactants. This is not consistent. The same treatment must be applied for both, reactants r and products p.

R4R2) The section which is related with this comment is corrected in the manuscript as:

The heat generation rate by the body was calculated by using equation (6), by following the same procedure as explained by Ulu et al [10]:

                                                                                         (6)

Where,  is the heat released by the body,  refers to the mole number rates of the products exiting the system and  is the mole number rate of the reactants entering the system;  represent the enthalpy of formation of the reactants  up taken at the standard conditions, and  is the enthalpy of formation of the products of the metabolism leaving the body at 37 oC. Values of   and  are presented in Table 4. and adapted from Kuddusi  [41].

C5R2) The format of the figures 1 and 3 is very poor. Increase quality and make a more appropriate figure format, e.g. y-axis: make 5 kg steps, no figure header (there is already a figure caption)

R5R2) Figures are redrawn to improve the quality

C6R2) Please revise equation 6 and check the results on changes. Then I can do a follow up review.

R6R2) Section related with Equation (6) has been rewritten as:

The heat generation rate by the body was calculated by using equation (6), by following the same procedure as explained by Ulu et al [10]:

                                                                                         (6)

Where,  is the heat released by the body,  refers to the mole number rates of the products exiting the system and  is the mole number rate of the reactants entering the system;  represent the enthalpy of formation of the reactants  up taken at the standard conditions, and  is the enthalpy of formation of the products of the metabolism leaving the body at 37 oC. Values of   and  are presented in Table 4. and adapted from Kuddusi  [41].

Round 3

Reviewer 2 Report

I have had two comments that are ignored by the authors: It is on the definition of standard enthalpies vs. enthalpies. I now understood that authors distinguish this as an enthalpy at a reference temperature (h0) and at any other temperature (h). However, this is wrong. Standard state means that all components are related to a chosen standard state which is infinite dilution or pure component. I adviced the authors on two articles that describe this:

https://doi.org/10.1038/s41598-021-85594-8

and

https://doi.org/10.1146/annurev-chembioeng-080615-034704

These works show that the standard state (all components at infinite dilution for biomolecules) differs from non-standard state by concentration effects of the metabolites or from solute(e.g., NaCl) interactions with the metabolites of the considered reaction.

--> 1) This has to be explained in the manuscript, and the works have to be cited.

--> 2) The use of standard has to be introduced for all enthalpies of this work. Concentration effects were not considered, thus all enthalpies are h0, some h0(T=298K) others are h0(T=310K).

It is very important that WE do this important thing correctly and do not confuse standard state just with a refernce temperature.

Author Response

RESPONSE TO THE COMMENTS OF REVIEWER 2

R2C1) I have had two comments that are ignored by the authors: It is on the definition of standard enthalpies vs. enthalpies. I now understood that authors distinguish this as an enthalpy at a reference temperature (h0) and at any other temperature (h). However, this is wrong. Standard state means that all components are related to a chosen standard state, which is infinite dilution or pure component. I advised the authors on two articles that describe this:

https://doi.org/10.1038/s41598-021-85594-8

and

https://doi.org/10.1146/annurev-chembioeng-080615-034704

These works show that the standard state (all components at infinite dilution for biomolecules) differs from non-standard state by concentration effects of the metabolites or from solute (e.g., NaCl) interactions with the metabolites of the considered reaction.

--> 1) This has to be explained in the manuscript, and the works have to be cited.

--> 2) The use of standard has to be introduced for all enthalpies of this work. Concentration effects were not considered, thus all enthalpies are h0, some h0(T=298K) others are h0(T=310K). 

It is very important that WE do this important thing correctly and do not confuse standard state just with a reference temperature.

R2R1) We added the recommended studies to the list of the references and then incorporated the reviewers remarks to the manuscript as: 

In Table 4, concentration effects on the enthalpies were neglected; therefore, some error is involved into our calculations. Standard state means that all components are related to a chosen standard state, which is infinite dilution or pure component [74, 75]. Fortunately, enthalpy differences are employed in equation 6, this may help to cancel some of this error.

Round 4

Reviewer 2 Report

my concerns have been partly resolved.